



**Evaluation of aerosol optical depths and clear-sky radiative fluxes of the**
**CERES Edition 4.1 SYN1deg data product**
David W. Fillmore[1], David A. Rutan[2], Seiji Kato[3], Fred G. Rose[2],
and Thomas E. Caldwell[2]
[1]University Center for Atmospheric Research, Boulder, CO. 80307
[2]SSAI, Hampton, VA, 23666
[3]NASA Langley Research Center, Hampton, VA, 23666
_______________
*Corresponding author address:* David W Fillmore,
University Center for Atmospheric Research
P.O. Box 3000
Boulder, CO 80307
E-mail: david.w.fillmore@ucar.edu



Abstract
Aerosol optical depths (AOD) used for the Edition 4.1 Clouds and the Earth's Radiant
Energy System (CERES) Synoptic (SYN1deg) are evaluated. AODs are derived from
Moderate Resolution Imaging Spectroradiometer (MODIS) observations and assimilated
by an aerosol transport model (MATCH). As a consequence, clear-sky AODs closely
match with those derived from MODIS instruments. AODs under all-sky conditions are
larger than AODs under clear-sky conditions, which is supported by ground-based
AERONET observations. When all-sky MATCH AODs are compared with Modern-Era
Retrospective Analysis for Research and Applications (MERRA2) AODs, MATCH
AODs are generally larger than MERRA2 AODS especially over convective regions (e.g.
Amazon, central Africa, and eastern Asia). The difference is largely caused by MODIS
AODs used for assimilation. Including AODs with larger retrieval uncertainty makes
AODs over the convective regions larger. When AODs are used for clear-sky irradiance
computations and computed downward shortwave irradiances are compared with ground-
based observations, the computed instantaneous irradiances are 1% to 2% larger than
observed irradiances. The comparison of top-of-atmosphere clear-sky irradiances with
those derived from CERES observations suggests that AODs used for surface radiation
observation sites are larger by 0.01 to 0.03, which is within the uncertainty of
instantaneous MODIS AODs. However, the comparison with AERONET AOD suggests
AODs used for computations over desert sites are 0.08 larger. The cause of positive
biases of downward shortwave irradiance and AODs for the desert sites are unknown.


## 1. __Introduction__

Accurate estimates of the radiative effects of clouds and aerosols are essential for an understanding the radiative forcing to the Earth's climate system (Bauer and Menon, 2012, Boucher et al. 2013). In addition, through the reflection and absorption of solar radiation, and the absorption and emission of terrestrial thermal radiation, clouds and aerosols affect the radiative heating of both the atmosphere and the surface, which in turn governs the atmospheric circulation and the hydrological cycle (e.g. Stephens et al. 2020, L'Ecuyer et al. 2015). Under the Earth Observing System (EOS) program, the National Aeronautics and Space Administration (NASA) has placed into orbit a series of satellites devoted to long term observations of the climate state. Among these are Terra and Aqua, the flagship satellites of the EOS. Central to observation of climate evolution are Moderate Resolution Imaging Spectroradiometer (MODIS) and the Clouds and the Earth's Radiant Energy System (CERES) instrument pairs that fly on both the Terra (March 2000 - present) and the Aqua (July 2002 - present) platforms (Wielicki et al. 1996). Additional CERES instruments were launched (October 2011) upon the Suomi National Polar-orbiting Partnership (NPP) satellite along with the MODIS successor, the Visible Infrared Imager Radiometer Suite (VIIRS), and on the NOAA-20 satellite (November 2017). In addition to observations from these satellites, the CERES mission also integrates observations from the Geostationary Operational Environmental Satellites (GOES) (West and East), as well as other geostationary satellites around the globe, for full diurnal coverage of clouds and radiation.

The CERES instruments measure broadband radiances over the solar spectrum (shortwave), the thermal infrared (longwave radiance is obtained from a total channel



minus the shortwave channel), and the near infrared atmospheric window, with frequent
on-board calibration. CERES measurements, in conjunction with MODIS information,
are used to infer broadband irradiances through empirical angular distribution models
(ADMs). Geosynchronous satellite imagery observes the diurnal cycle of clouds, which is
not fully sampled by the polar orbiting satellites upon which CERES and MODIS reside.
While top-of-atmosphere (TOA) irradiances are derived from broadband
radiances measured by CERES instruments (Loeb et al. 2005; Su et al. 2015), surface and
in atmosphere irradiances are computed with a radiative transfer model. Inputs used for
the computations include cloud properties derived from MODIS and geostationary
satellites, aerosol optical depth derived from MODIS radiances, and surface albedo
derived from MODIS and CERES observations (Rutan et al. 2009). Temperature and
humidity profiles are provided by a reanalysis product produced by the NASA Goddard
Modeling and Assimilation Office (GMAO).
Irradiances at the surface produced by the CERES team have been compared with
surface observations (Rutan et al. 2015; Kato et al. 2013, 2018). These comparisons are
for all-sky conditions (i.e. including any clouds). Irradiances under clear-sky conditions
are not explicitly separated from all-sky conditions in the evaluations. There are several
reasons that impede efforts at rigorous validation of clear-sky irradiances with surface
observations; 1) a clear-sky condition at a given site does not persist over a long time
(e.g. a month or longer), 2) there are mismatches of clear-sky conditions determined by
satellite- and ground-based instruments, and 3) field-of-view size between CERES
instruments and ground-based radiometers differ.



Despite these difficulties for evaluating computed clear-sky irradiances, clear-sky

irradiances play an important role in quantifying aerosol and cloud radiative effects (Loeb
and Su 2010; Soden and Chung 2017). Therefore, the uncertainty in surface irradiances
need to be understood in order to assess the uncertainty in aerosol and cloud radiative
effect. This work is the first attempt by the CERES team to evaluate clear-sky surface
irradiances provided by its data products. One of the essential variables in computing
clear-sky irradiances is aerosol optical depth. In this paper, we evaluate aerosol optical
depth used for irradiance computations in the CERES project and analyze how the error
propagates to clear-sky surface irradiances. Computations of surface irradiances provided
by Edition 4.1 SYN1deg data products use aerosol optical depth derived by a chemical
transport model [The Model for Atmospheric Transport and Chemistry (MATCH, Collins
et al. 2001)] that assimilates MODIS-derived aerosol optical depth. The MATCH model
is described in Section 1. In Section 2, we explain the aerosol transport model briefly. In
Section 3, the assimilation of aerosol optical depth in the model is discussed. Sections 4
and 5 compares aerosol optical depths used by the CERES team with, those from
MERRA2 and the Aerosol Robotic Network (AERONET, Holben et al. 1998).

**2.  Description of MATCH model**

The Model for Atmospheric Transport and Chemistry (MATCH) is a transport

model of intermediate complexity driven by offline meteorological fields from the
National Centers for Environmental Prediction (NCEP) reanalysis. It is run on a $194 \times 96$
($1.9° \times 1.9°$) spatial grid with a vertical resolution of 28 sigma-p levels. Temporally, the
meteorological fields are linearly interpolated to 30-minute times at which time the





chemical processes are run. One exception is that the sulfur model is interpolated again to
run at 2-min subscale time steps. MATCH is one of the many aerosol transport models
that participated in the AeroCom model inter-comparison project (Textor et al., 2006;
Kinne et al. 2006; Textor et al. 2007) and the AeroCom carbon inter-comparison project
(Koch et al., 2009; Huneeus et al., 2011).

Aerosol types included in MATCH are small dust, large dust, sulfate, sea salt,

soot, soluble particles, and insoluble particles (**Table 1**). Model physics included in
MATCH are parameterizations for convection and boundary layer processes, along with
prognostic cloud and precipitation schemes for aqueous chemistry and the scavenging of
soluble species. MATCH also includes the ability to resolve the transport of aerosols via
convection, boundary layer transport, and scavenging and deposition of soluble gases and
aerosols. MATCH can simulate most cloud processes currently in use in a GCM (eg.
cloud fraction, cloud water and ice content, fraction of water converted to rain and snow,
and evaporation of condensate and precipitate). It also includes vertical turbulent eddy
processes. These processes are then used for convective transport, wet scavenging, wet
deposition and dry deposition of the MATCH aerosols. These various parameterizations
were developed, originally, for the NCAR Community Climate Model (CCM) and
subsequently incorporated into the MATCH model. Descriptions of these
parameterizations are given by Rasch et. al (1997, 2001), Collins et. al (2001) and
additional papers described therein.

The MATCH aerosol suite includes a detailed mineral dust scheme in the Dust

Entrainment and Deposition model, (Zender et al., 2003), and a diagnostic
parameterization for sea-salt aerosol based on the 10m wind speed (Blanchard and



Woodcock, 1980). The sulfur cycle and the chemical reactions for sulfate aerosol creation
rely on monthly climatological oxidant fields and emission inventories (**Table 1**) for
sulfur oxides and oceanic dimethyl sulfide (photochemistry and nitrate aerosol are
omitted). The reaction scheme is similar to that of the Model for Ozone and Related
Chemical Tracers (MOZART), (Emmons et al., 2010).  Carbon aerosols (both organic
compounds and soot) evolve with simple mean lifetime e-foldings from surface fluxes
specified through natural, biomass burning and fossil fuel burning emission inventories
(also monthly climatologies given in **Table 1**).

Table 1. Aerosol Types & Climatological Sources

| Aerosol Type | Source | Description |
|---|---|---|
| Sea Salt | Blanchard and Woodcock, 1980 | Wind Driven |
| Dust | Ginoux et al. (2001); Zender et al. (2003) | NCEP soil moisture, wind driven |
| Sulfate (natural & anthropogenic) | Benkovitz et al. (1996); Barth et al. (2000) | monthly climatological |
| Carbon (organic & Soot) | Liousse et al. (1996) | monthly climatological |
| Volcanic | Episodic inclusion of Sulfur dioxide | Processed by model |
| | | |


The optical properties of the various aerosol types (e.g. mass extinction

coefficient, single scatter albedo), which are key parameters for aerosol assimilation, are
drawn from the standard Optical Properties of Clouds and Aerosols (OPAC, Hess et al.
1998) database. Scattering properties from MATCH are not used directly in the radiative
transfer calculations in the SYN1deg. Instead, aerosol types from MATCH are mapped to
a similar set of scattering properties embedded in the Langley Fu & Liou radiative
transfer code (Fu and Liou, 1993; Fu et. al 1998; Rose et. al 2013). These include OPAC



as in MATCH for all but the small and large dust particles. Dust scattering and absorption
properties in the Langley Fu & Liou code are from Sinyuk et al. (2003).

**2.1 MATCH Assimilation of MODIS Aerosol Optical Depths**

One major advantage of the MATCH model is its ability to reliably assimilate

satellite-based retrievals of aerosol optical depth (AOD) to constrain the climatologically
forced aerosols generated within the chemical transport portion of the code.  Edition 4
MATCH algorithms ingest MODIS Collection 6.1 AOD (Remer et al., 2005), beginning
in March 2000 from the Terra satellite and June 2002 from both Terra and Aqua
satellites. The MATCH assimilates MODIS AOD at the green wavelength of 550 nm.
MATCH combines AOD derived by the Dark Target (Levy et al. 2013) and Deep Blue
algorithms (Hsu et al., 2006). A global daily mean AOD in a 1.9°x1.9° grid is derived
from Terra and Aqua observations by simply averaging available Terra and Aqua dark
target and deep blue derived AODs in a grid box. Unlike dark target and deep blue
merged product (MOD08), we do not use a quality assurance confidence (QAC) score to
screen AOD.

Because Terra and Aqua overpass time is 10:30 AM and 1:30 PM local time,

AOD at local solar noon is assimilated by taking a 15° longitude width of retrieved AOD
from the daily mean map.  Examples of the magnitude of AOD adjustments by the
assimilation are shown in **Fig. 1. Figure 1a** shows hourly AOD field differences, 4 UT
minus 3 UT on February 1st, 2020. Similarly, **Figure 1b** shows 10 UT minus 9 UT of the
same day. The 15° vertical band is clearly visible where red (blue) colors indicate total
column aerosol is increased (decreased) by the MODIS AOD assimilation. Following the



AOD adjustment, aerosol masses in the atmospheric column through the troposphere are
scaled to closely match the AOD derived from MODIS. Neither the vertical profile nor
the relative abundance of the aerosol species is adjusted. Once aerosol mass is adjusted at
the local noon for the regions where MODIS AOD is available, the adjusted aerosol mass
is carried on to the next time step. Besides the MODIS adjustments, wind driven sea-salt
creation and deposition are found along frontal boundaries in the North Atlantic and
Southern Oceans. The maps also indicate hourly increases and decreases in high aerosol
loading areas such as those found around China and SE Asia.

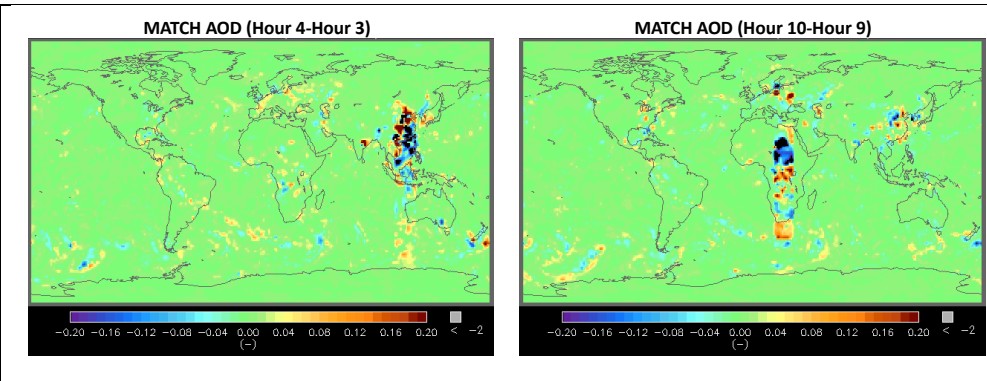

Figure 1. Difference of MATCH AOD due to the assimilation of MODIS AOD. The left plot is 4 UT minus 3 UT and right plot is 10 UT minus 9 UT on February 1, 2020. AOD is adjusted at the local solar noon within the 15° longitudinal band by the MODIS AOD assimilation. Wind-blown dust and sea salts differences are also apparent outside the 15° longitudinal band.


Episodic events such as intense fires or volcanic eruptions are not specifically

included in the MATCH aerosol package. Such events are captured by the assimilation of
MODIS AOD and total column aerosol loading is adjusted upward. The adjustment is
applied to AOD only. The aerosol type (and so scattering properties) is not adjusted to
reflect the reality of the scattering or absorbing aerosol during such an event.


## 2.2 MATCH and MERRA2 comparison


In this section, we compare AODs between MATCH and MERRA2 (Randles et
al., 2017) in which MODIS clear-sky radiances are assimilated. MERRA2 also
assimilates surface observed AOD by AEROENT and ship born AOD observations. We
compare AODs in two different ways. First, MATCH and MERRA2 AODs are compared
with MODIS AODs. The first comparison tests the consistency of daily means when
MODIS aerosol optical depth is available (i.e. clear somewhere in the grid box at Terra
and Aqua overpass time). Second, MATCH and MERRA2 AODs are compared under
all-sky conditions, which is only possible with modeled AODs.

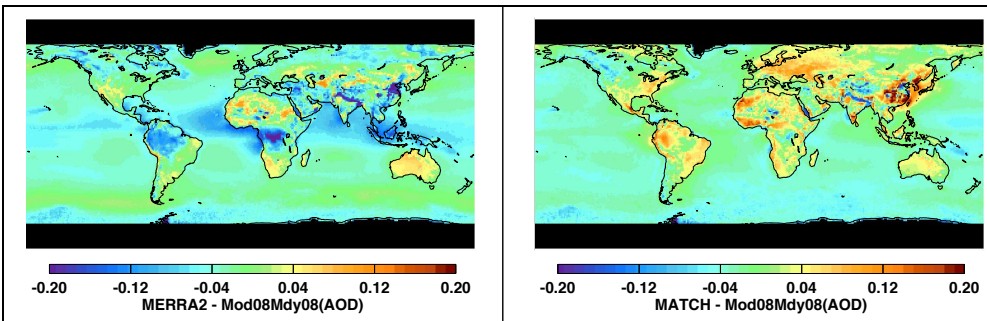

**Figure 2**: Monthly mean aerosol optical thickens (AOD, i.e. $\langle AOD_{MODIS}^{clr} \rangle$ see texts for the definition) difference of left) MERRA2 – MODIS and right) MATCH – MODIS averaged over Mar 2000 through Feb 2020. MERRA2 and MATCH daily mean AODs are sampled when daily mean MODIS AOD from the same 1°×1° grid is available. Sampled daily mean AODs are subsequently averaged. MODIS AODs are 1°×1° average of daily mean AODs derived by the dark target and deep blue algorithms using Terra and Aqua observations.


**Figure 2** shows differences of monthly mean AOD between MERRA2 and
MODIS on the left and MATCH and MODIS on the right. To compute the monthly mean
AOD differences, both MERRA2 and MATCH daily mean AODs are sampled when
daily mean MODIS AOD from the same 1°×1° grid is available (hereinafter $AOD_{MODIS}^{clr}$).


Sampled daily mean AODs ( $AOD_{MODIS}^{clr}$ ) are subsequently averaged (hereinafter
$\langle AOD_{MODIS}^{clr} \rangle$, where the bracket indicates a simple arithmetic monthly mean). Although
both products assimilate MODIS observations, each shows fairly significant differences
from MODIS values. Differences arise because MODIS daily mean AOD is clear-sky
AOD at Terra and Aqua overpass time only while MERRA2 and MATCH daily mean
AOD includes AOD from other times of the day. When the non-overpass time is also
clear, MATCH $AOD_{MODIS}^{clr}$ should be close to MODIS $AOD_{MODIS}^{clr}$. However, when clouds
are present in MATCH during non-overpass times, modeled AOD are used, hence the
daily mean AOD can deviate from MODIS $AOD_{MODIS}^{clr}$. In addition, AOD differences for
MERRA2 at Terra and Aqua overpass times might be larger than MATCH even for clear-
sky conditions as MERRA2 assimilates observed AOD data other than MODIS AOD.

While MATCH shows large positive differences over land, especially China and

south east Asia, Australia, Amazon, and north Africa, MERRA2 shows significantly
negative differences over major rain-forest regions of south America, Africa, and the
tropical western Pacific. Both $\langle AOD_{MODIS}^{clr} \rangle$ are closer to MODIS AOD over ocean
compared to $\langle AOD_{MODIS}^{clr} \rangle$ over land except MERRA2 shows a negative difference across
the Indian ocean and off the west coast of Africa in the Atlantic Ocean. When MODIS
$AOD_{MODIS}^{clr}$ is available in the grid box, MATCH weighs MODIS AOD heavily in
assimilating MODIS AOD at local solar noon so that MATCH AOD is nearly identical to
MODIS AOD at the local noon under clear-sky regions. As a consequence, the difference
of global monthly mean MATCH and MODIS $AOD_{MODIS}^{clr}$ is smaller than the difference
of MERRA2 and MODIS $AOD_{MODIS}^{clr}$.
**Figure 3** shows the difference of $AOD_{MODIS}^{clr}$ more clearly. In **Fig. 3** $AOD_{MODIS}^{clr}$ are
compared directly in a log-density plot where each point represents a comparison for the
daily average of a given grid box; MERRA2 versus MODIS on the left and MATCH
versus MODIS on the right. **Figure 3** indicates that MATCH $AOD_{MODIS}^{clr}$ has a smaller
bias with respect to the MODIS AOD than the MERRA2 AOD but has approximately the
same RMS compared to the MERRA2 $AOD_{MODIS}^{clr}$.

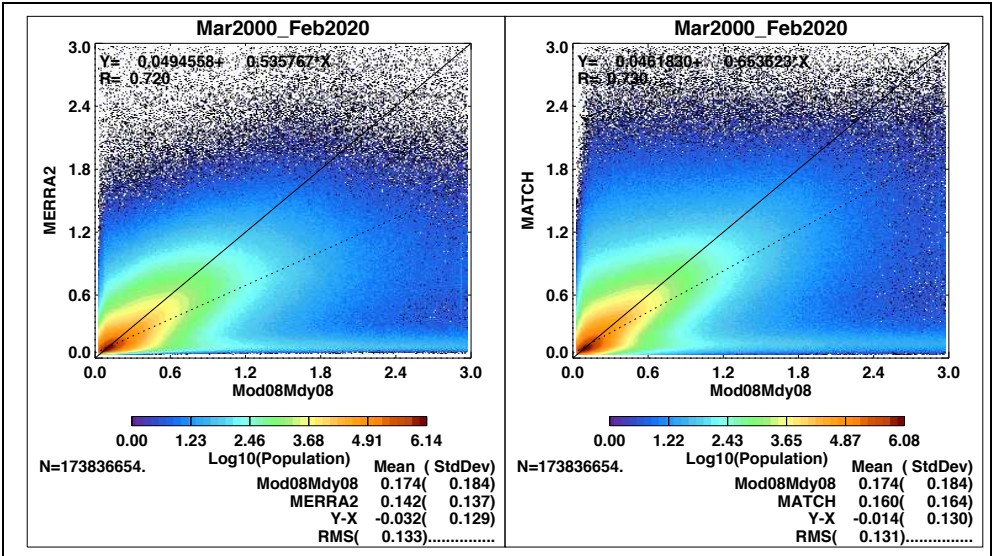

**Figure 3**: Scatter plot of daily 1°×1° mean aerosol optical depth (AOD) from a) MERRA2 and b) MATCH versus AOD derived from MODIS on Terra and Aqua for Mar 2000 through Feb 2020. MODIS AODs are 1°×1° daily average of AODs derived by the dark target and deep blue algorithms. Only days and grid boxes that have MODIS AOD (i.e. $AOD_{MODIS}^{clr}$ defined in the texts) are used.

**Figure 4** shows 1°×1° monthly mean maps of MATCH AOD on the left and its
difference from MERRA-2 on the right for all sky (top maps) and clear sky (bottom
maps) conditions for March 2000 through February 2020. The clear-sky monthly mean
aerosol optical depth is derived by averaging daily mean aerosol optical depth weighted





by clear fraction (hereinafter $\overline{AOD_{MODIS}^{clr}}$, overbar indicates monthly mean), where the
clear fraction is derived from MODIS on Terra and Aqua (Minnis et al. 2020). MATCH
all-sky AOD (hereinafter $\overline{AOD^{all}}$) is larger than MERRA2 $\overline{AOD^{all}}$), particularly over the
rain forest regions of the globe as well as India and China. Although the difference is
smaller, the difference of $\overline{AOD_{MODIS}^{clr}}$ shows a similar spatial pattern (**Fig. 4** bottom right)
to the all-sky difference. This is consistent with **Fig. 2**, showing that MERRA2 tended to
underestimate $AOD_{MODIS}^{clr}$ with respect to MODIS $AOD_{MODIS}^{clr}$. A larger difference over
convective regions (e.g. Amazon, central Africa, and south east Asia) is caused by how
dark target and deep blue AOD are merged. As mentioned earlier, we do not use QAC to
screen AOD. Convective clouds introduce a larger uncertainty to AOD because of a 3D
radiation effect or poor fit to observations with retrieved AOD (personal communication
with R. Levy 2020). For these situations, AODs associated with QA confidence scores
less than 2 are screened out in the MOD08 dark target and deep blue merged product
(Levy et al. 2013).

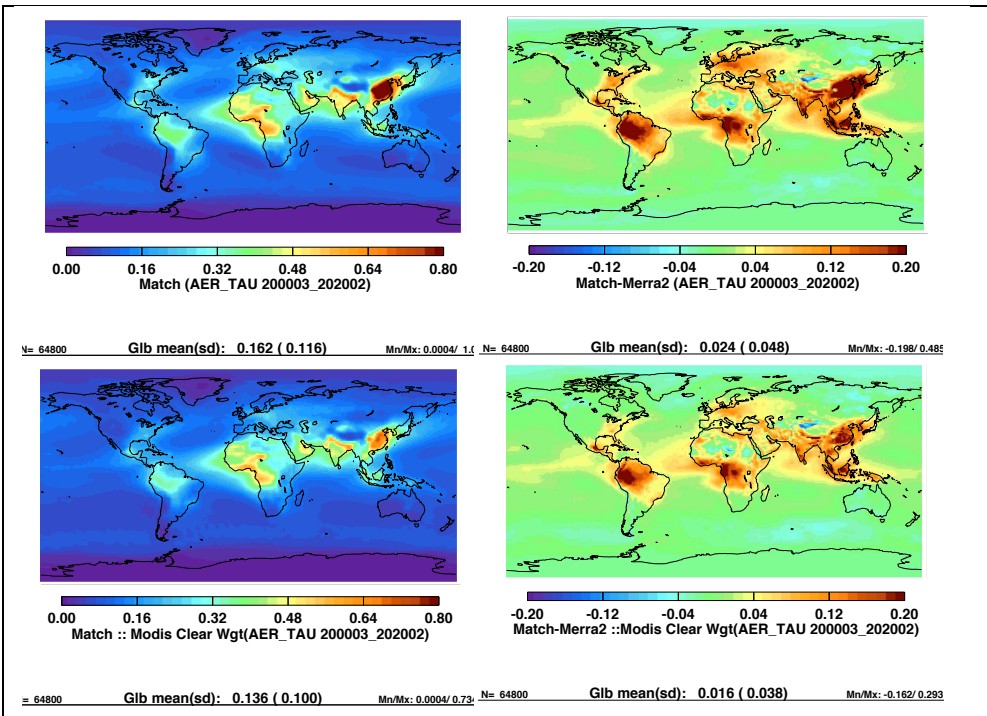

**Figure 4**. Left) Monthly mean aerosol optical depth (AOD) from MATCH and right) the difference between MATCH and MERRA2 (MATCH – MERRA2) for January 2020. Top maps are for all-sky (i.e. $AOD_{MODIS}^{all}$, bottom maps are for clear-sky (i.e. $AOD_{MODIS}^{clr}$). Clear-sky monthly mean aerosol optical depth is derived by averaging daily mean aerosol optical depth weighted by daily 1°×1° gridded mean clear fraction where the clear fraction is derived from MODIS on Terra and Aqua.


**2.3 Comparison with AERONET**
Above results indicate that both MATCH and MERRA2 $\overline{AOD_{MODIS}^{clr}}$ are
generally, respectively, larger and smaller than MODIS $\overline{AOD_{MODIS}^{clr}}$ . Larger difference
between MATCH and MERRA2 $\overline{AOD^{all}}$ over convective regions originated from
merged AOD product used for the assimilation. Of primary importance to radiative
transfer calculations within the SYN1deg product is the ability of the MATCH model to
accurately represent total column aerosol optical depth. To test the overall accuracy, we
use observations from the AERosol RObotic NETwork (AERONET).  AERONET is a





global federation of ground-based remotes sensing sites developed by NASA and now
supported by a number of institutions around the world (Holben et al. 1998). Each site
maintains a CIMEL sun-photometer that scans the daytime sky every 20 minutes.
Collected data are processed according to standards of calibration and processing
maintained by the AERONET project. Here we utilize Level 2.0, data that have been
screened for clouds and quality assured (Smirnov et al. 2000).

**Figure 5** shows an hourly time series of AOD from MATCH, MERRA2 and

AERONET for January 2010 at the Beijing China AERONET site. The top plot shows
cloud fraction time series derived from MODIS and GEOs from the SYN1deg Ed4.1
product (Rutan et al. 2015), and the bottom plot shows AOD time series. Generally, both
models produce a large variability of AOD at this site fairly well over the course of the
month.  While both MERRA2 and MATCH AODs increase near times when cloud
fraction approaches 100%, the increase of MATCH AOD, which correlates with the
increase of AERONET AOD relatively well, is larger than the increase of MERRA2
AOD. Although the temporal correlation coefficient of the MATCH and AERONET
AODs is smaller at this site during summer months than during winter months (not
shown), a good temporal correlation between MATCH and AERONET AODs is
consistent across most locations and times we considered. To show this statistically, in
the following, we extend this analysis to a number of AERONET sites grouped
geographically based on general aerosol type.

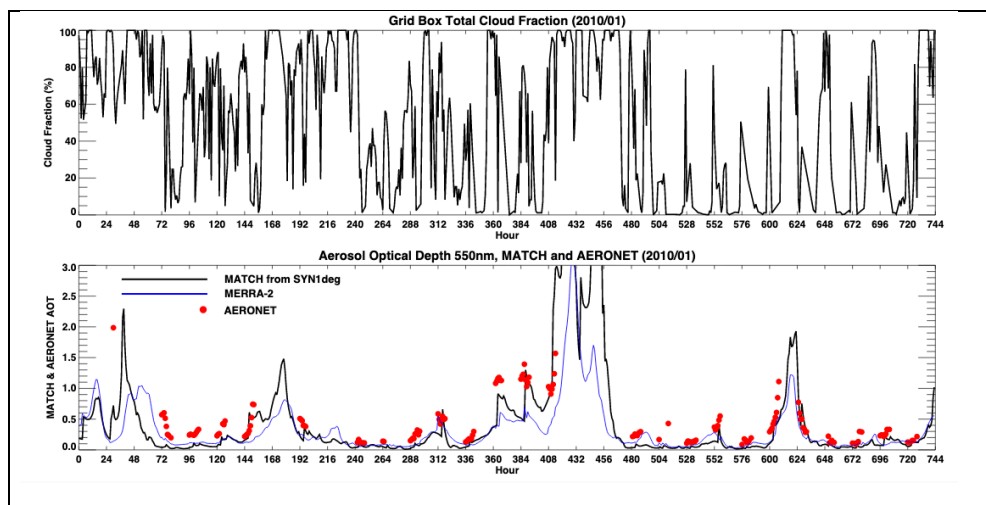

**Figure 5**. Hourly time series of grid box cloud fraction (top) from SYN1deg Ed4.1 CERES product and Aerosol Optical Depths (bottom). Results from the grid box containing the AERONET Beijing, CH site. Black line MATCH, blue line MERRA-2, red dots, AERONET observations. MATCH and, to a lesser degree MERRA-2 often have large increases in AOD when cloud fraction nears 100%.


Aerosol optical depths from AERONET are nominally provided at 8 spectral
channels, every 20 minutes given favorable conditions. We use two channels to derive
observed AOD at 550 nm to compare to the AOD provided by the MATCH model.
Because the SYN1deg radiative transfer calculation is done hourly, we average any
observations within a given hour period centered at the $30^{th}$ minute for each site
collocated within a SYN1deg grid box. AERONET sites chosen are shown in **Fig. 6** with
a complete listing of all sites in Appendix 1. Though we examine 45 sites over 20+ years,
we aggregate the statistics within continental regions which naturally isolates them by
general climatic conditions. Tables 2 and 3 show comparisons for each site grouping,
respectively, for clear sky (less than 1% cloud identified by MODIS and geostationary
satellites in the SYN1deg grid box) conditions and for all sky (any cloud condition within



the SYN1deg grid box) conditions. Using clear-sky scenes identified by MODIS only
gives the same statistical results with fewer number of samples. Statistics shown in
Tables 2 and 3 are the average observed value, mean bias (MATCH – Observation), root
mean square (RMS) difference and the correlation coefficient (R) over the time period
from March 2000 through February 2020. The actual time period varies depending on the
site due to AERONET data availability. The RMS difference and correlation coefficient
are computed by each site with hourly mean values where observations are available
from March 2000 through February 2020. For comparison purposes we show the same
statistics derived from observations compared to MERRA2 AODs using the identical
hours. We note, however, that MERRA2 assimilates AERONET while MATCH AODs
are independent from AERONET AODs.

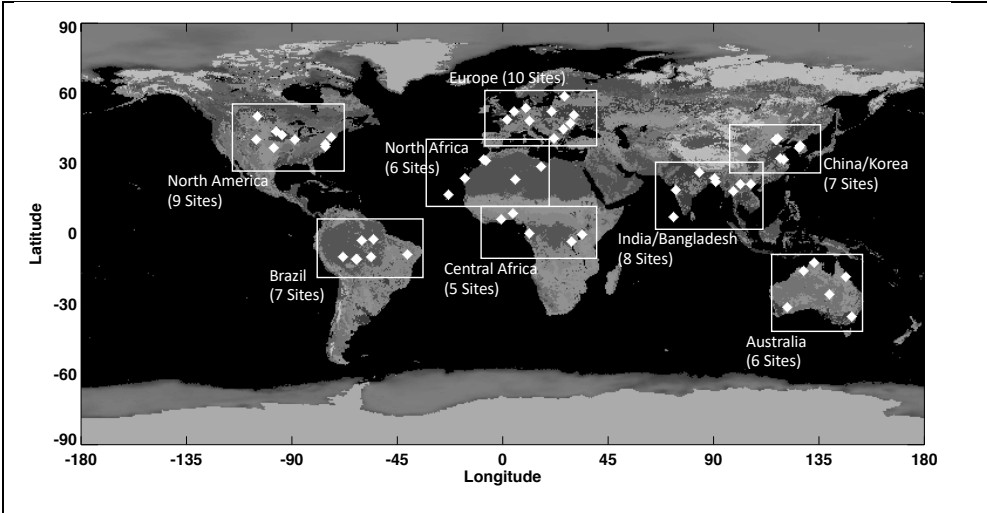

**Figure 6**. Location of AERONET sites and how they grouped for calculations of mean/bias/RMS with respect to MATCH and MERRA-2 optical depths found in tables 2 and 3.


MATCH AOD for the Brazil group is biased high by 0.03, and China south east

Asia has no bias compared with AREONET AODs. These two regions have relatively





large bias of $\langle AOD_{MODIS}^{clr} \rangle$ from MATCH compared with MODIS AODs (**Fig. 2** right). In
contrast, negative bias of MERRA2 AODs compared with AERONET AODs for Brazil,
central Africa, and China/South East Asia groups are consistent with negative bias of
MERRA2 $\langle AOD_{MODIS}^{clr} \rangle$ compared with MODIS AODs (**Fig. 2** left). For the China/south
east Asia group, the RMS difference between MATCH AODs and AERONET AODs is
0.18 and correlation coefficient is 0.7. These are worse than the counterpart values of
MERRA2 versus AERONET AODs because summertime agreement between MATCH
and AEROENT AODs is worse if a similar plot as **Fig. 5** is plotted for summertime when
hygroscopic aerosols are dominant under high relative humidity conditions.

The sign of the MATCH AODs compared to AERONET AODs for all-sky

conditions is generally consistent with the sign of clear-sky counterparts. The RMS
difference under all-sky conditions is generally larger than the clear-sky RMS difference
while the correlation coefficient is nearly the same. The biases for MERRA2
comparisons are generally comparable to MATCH though RMS for MERRA2 tend to be
slightly smaller and correlations tend to be higher due in part to the assimilation of
AERONET into the MERRA2 model.











Table 2. Hourly AERONET station statistics for MATCH and MERRA-2.
Continental Groups, Clear Sky conditions[1]

| Site | Predominant Aerosol Type | Number | Observed Average | MATCH | | | MERRA-2 | | |
| --- | --- | --- | --- | --- | --- | --- | --- | --- | --- |
| | | | | Bias | RMS | $R^2$ | Bias | RMS | $R^2$ |
| *Australia (5 Sites)* | Dust Smoke | 20925 | 0.06 | 0.01 | 0.06 | 0.4 | 0.03 | 0.05 | 0.7 |
| *Brazil (7 Sites)* | Smoke Polluted | 6554 | 0.14 | 0.02 | 0.10 | 0.8 | -0.02 | 0.08 | 0.9 |
| *Central Africa (5 Sites)* | Smoke | 2139 | 0.70 | -0.10 | 0.24 | 0.9 | -0.10 | 0.24 | 0.9 |
| *North Africa (5 Sites)* | Dust | 10047 | 0.17 | 0.07 | 0.15 | 0.7 | 0.02 | 0.10 | 0.8 |
| *China SE Asia (8 Sites)* | Polluted | 2827 | 0.26 | -0.00 | 0.18 | 0.7 | -0.03 | 0.15 | 0.8 |
| *India/Bangledesh (6 Sites)* | Smoke Polluted | 3010 | 0.51 | -0.09 | 0.28 | 0.6 | -0.10 | 0.24 | 0.8 |
| *North America (9 SItes)* | Continental Polluted | 21429 | 0.10 | -0.00 | 0.07 | 0.7 | 0.00 | 0.06 | 0.8 |
| *Europe (10 Sites)* | Continental Polluted | 10211 | 0.13 | 0.01 | 0.07 | 0.7 | -0.02 | 0.05 | 0.8 |
| | [1]The time period used is from Mar 2000 through Apr 2020. Actual period varies by site depending on AERONET data availability. Clear Sky is identified by MODIS and geostationary satellites and the cloud fraction is less than 1% over a SYN1deg grid box. | | | | | | | | |

















Table 3. Hourly AERONET station statistics for MATCH and MERRA-2.
Continental Groups, All Sky Conditions[1]

| Site | Predominant Aerosol Type | Number | Observed Average | MATCH | | | MERRA-2 | | |
|---|---|---|---|---|---|---|---|---|---|
| | | | | Bias | RMS | R[2] | Bias | RMS | R[2] |
| Australia (5 Sites) | Dust Smoke | 110523 | 0.09 | 0.00 | 0.09 | 0.5 | 0.02 | 0.07 | 0.8 |
| Brazil (7 Sites) | Smoke Polluted | 72656 | 0.25 | 0.03 | 0.23 | 0.8 | -0.04 | 0.18 | 0.9 |
| Central Africa (5 Sites) | Smoke | 41193 | 0.55 | -0.07 | 0.26 | 0.8 | -0.10 | 0.26 | 0.9 |
| North Africa (5 Sites) | Dust | 43205 | 0.23 | 0.08 | 0.20 | 0.7 | 0.01 | 0.14 | 0.8 |
| China SE Asia (8 Sites) | Polluted | 52287 | 0.45 | 0.01 | 0.31 | 0.7 | -0.08 | 0.27 | 0.8 |
| India/Bangladesh (6 Sites) | Smoke Polluted | 44534 | 0.61 | -0.06 | 0.32 | 0.6 | -0.10 | 0.32 | 0.7 |
| North America (9 SItes) | Continental Polluted | 160356 | 0.13 | 0.02 | 0.13 | 0.6 | 0.00 | 0.09 | 0.7 |
| Europe (10 Sites) | Continental Polluted | 175010 | 0.18 | 0.04 | 0.14 | 0.6 | -0.02 | 0.08 | 0.8 |

[1] The time period used for the statistics is from March 2000 through April 2020. The actual period varies by site depending on AERONET data availability.
[2] Correlation coefficient.


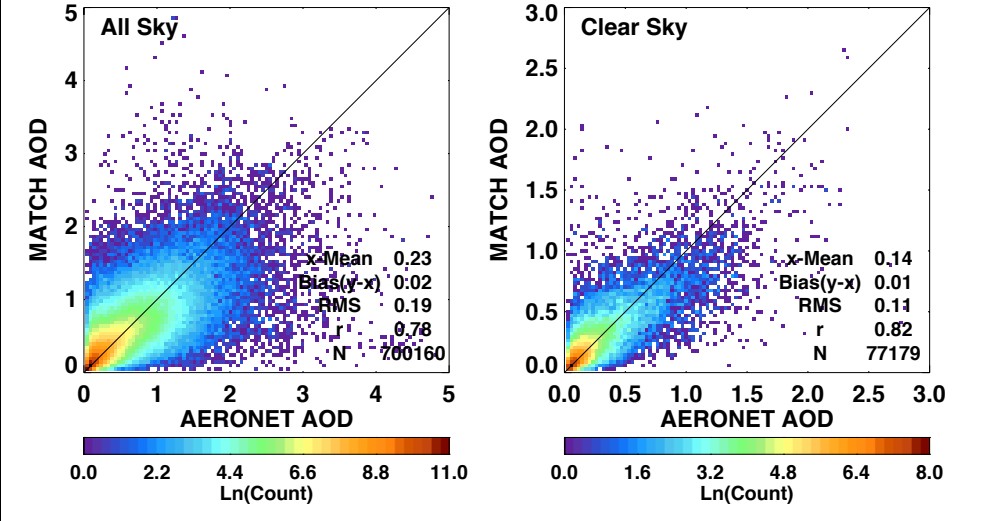

**Figure 7**. All sky (left) and clear sky (right) comparisons of observed (AERONET) hour mean optical depths to estimates from the MATCH model for 20+ years at 45 AERONET sites shown in **Fig 6**.




Results for all points across all sites and times are shown in **Fig. 7**. The color

density plots are in log scale and indicate the vast majority of observations have an AOD
of less than one for both clear and all sky conditions observed within the SYN1deg grid
box. Biases are less than 10% of the mean value but RMS is large relative to the mean
observed value. Overall correlation is approximately 0.8. The 'clear sky' hours (where
SYN1deg estimated less than 1% cloud in the grid box based on MODIS and GEO
observations) is a little more than 10% of the overall points. When MATCH AOD is
compared to MERRA2 AOD (not shown) MATCH is biased approximately 10% higher.

**3.        Discussion of AOD Differences**

In this section, we investigate the reason for the AOD differences shown in the

previous section. In addition, we estimate the effect of the AOD differences to surface
irradiances when MATCH AODs are used for surface irradiance computations.

Generally, cloud contamination in MODIS AODs is caused by unresolved sub-

pixel scale clouds (Kaufman et al. 2005; Martins et al. 2002). The difference shown over
convective regions, therefore, seems to be caused by the uncertainty due to 3D radiative
effects that impact retrieved AODs by unknown amounts (Wen et al. 2007), by errors in
estimating the fraction of hygroscopic aerosols or by the errors in estimating water uptake
by hygroscopic aerosols (Su et al 2008). Larger AODs are screened out in the MOD08
data product while the CERES team uses all retrieved AODs regardless of the QAC
score. The comparison with AERONET AODs is not decisive to determine how to screen
MODIS AODs because MATCH AODs are positively biased and MERRA2 AODs are
negatively biased for the Brazil group. The result underscores the difficulty of deriving



381 accurate AODs, which appear to involve requirements in addition to identification of

382 clear-sky scenes. Levy et al. (2013) list reasons lowering the QAC score as 1) pixels are

383 thrown out due to cloud masking, 2) retrieval solution does not fit the observation well,

384 and 3) the solution is not physically plausible given the observed situation. Therefore,

385 even though the difficulty of identifying clear-sky scenes is driven by cloud

386 contamination by trade cumulus (Loeb et al. 2018), the difficulty of deriving AODs exists

387 over convective regions (Varnai et al., 2017).

388  Larger positive biases of MATCH AODs compared with AERONET AODs exist

389 over Africa (Tables 2 and 3). For North Africa, the bias is known to be caused excessive

390 dust generated by the MATCH algorithm. Even though modeled aerosols are not often

391 used over north Africa owing to the abundance of clear-sky conditions, the dust problem

392 leads to a larger positive AOD bias. In addition, MATCH uses fixed aerosol sources in

393 time. Therefore, it tends to miss large aerosol events, such as forest fires, until clear-sky

394 conditions occur, allowing observations of the event by MODIS. This leads to a larger

395 RMS difference and lower correlation coefficient with AERONET AODs compared with

396 those from MERRA2 versus AERONET.

397  Because MODIS AOD are not generally available under overcast conditions, the

398 reliance on modeled AOD increases as the cloud fraction over a 1°×1° grid increases.

399 **Figure 5**, which shows that AERONET AOD increases with cloud fraction derived from

400 satellites, indicates that as the cloud fraction over a 1°×1° grid increases, AOD over the

401 clear-sky portion of the grid increases. In addition, **Fig. 5** suggests that modeled AODs

402 under near overcast conditions are significantly larger than clear-sky AODs that are

403 constrained by MODIS observations. Because we are not able to evaluate AODs under



overcast conditions, here we only assess AOD changes with cloud fraction using ground-
based observations. **Figure 8** shows the distribution of AERONET AODs for clear-sky
and all-sky conditions, as well as precipitable water derived from a microwave
radiometer separated by these two conditions. Clear-sky is identified by the Long-
Ackerman algorithm (Long et al. 2006) that uses surface direct and diffuse irradiances.
**Figure 8** shows that AOD and precipitable water under all-sky conditions are
significantly larger than those under clear-sky conditions. When we use cloud fraction
derived from satellite and plot AOD and precipitable water as a function of the cloud
fraction using the same grid box where the ground site is located, AOD and precipitable
water increase with the cloud fraction (**Fig. 9**). Therefore, increasing AOD with cloud
fraction shown in **Fig. 5** is qualitatively explained by increasing AOD of hygroscopic
aerosols with relative humidity. However, **Fig. 9** indicates that either the growth of
MATCH AOD seems to be too strong or modeled MATCH AOD under all-sky
conditions is too large.





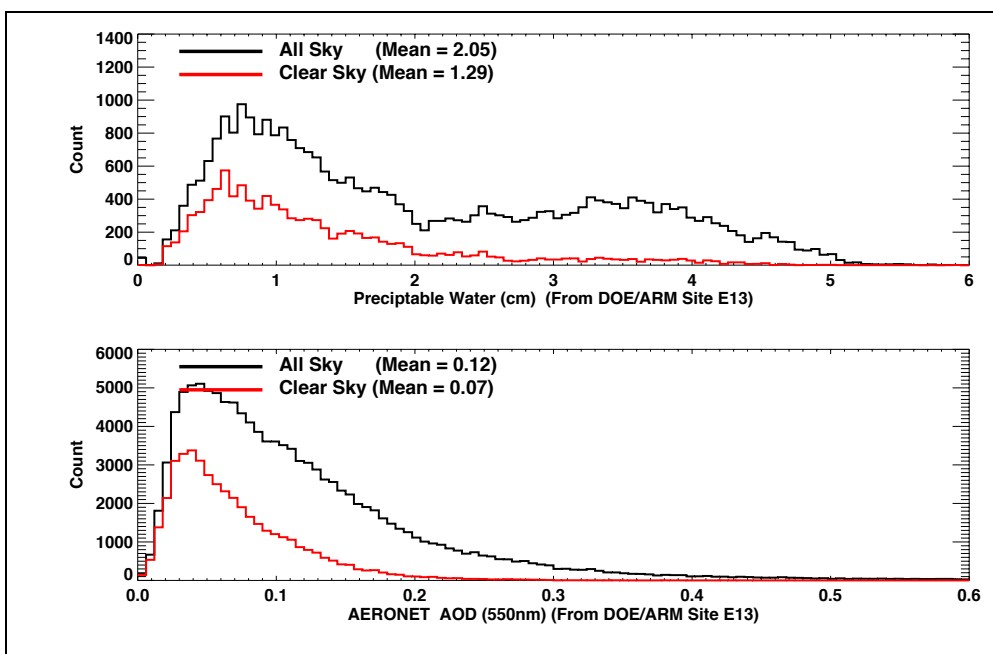

**Figure 8**. a) 15-minute mean precipitable water distributions from Microwave radiometer observations at ARM/SGP E13 site under all sky and clear sky conditions. b) 15-minute mean aerosol optical depth distributions from AERONET sun-photometer at 550nm. 'Clear sky' is here defined as when a 15-minute time period where the SWFA, surface radiometry-based cloud fraction, equals 0.






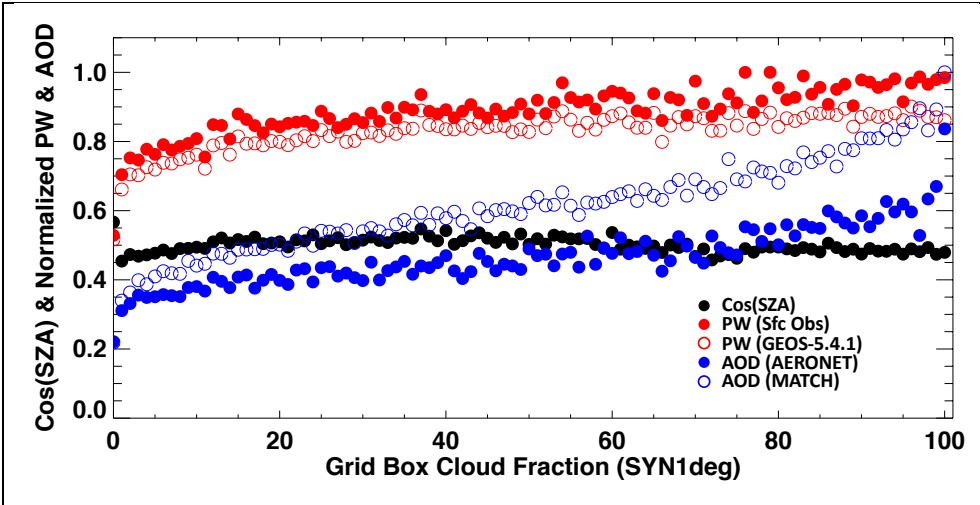

**Figure 9.** Aerosol optical depth (AOD) and precipitable water (PW) as a function of cloud fraction over the 1°×1° grid box where the ARM/SGP E13 and SURFRAD Bondville IL sites are located. Closed and open blue circles are, respectively, AOD derived from AERONET and MATCH AOD. Closed and open red circles are, respectively, PW derived from microwave radiometer and CIMEL sun photometer and GEOS-5.4.1 PW. Cloud fractions are derived from MODIS and geostationary satellites. Black dots are mean cosine solar zenith angle of the time of AOD and PW observations. AOD and PW are normalized to their maximum value for display.







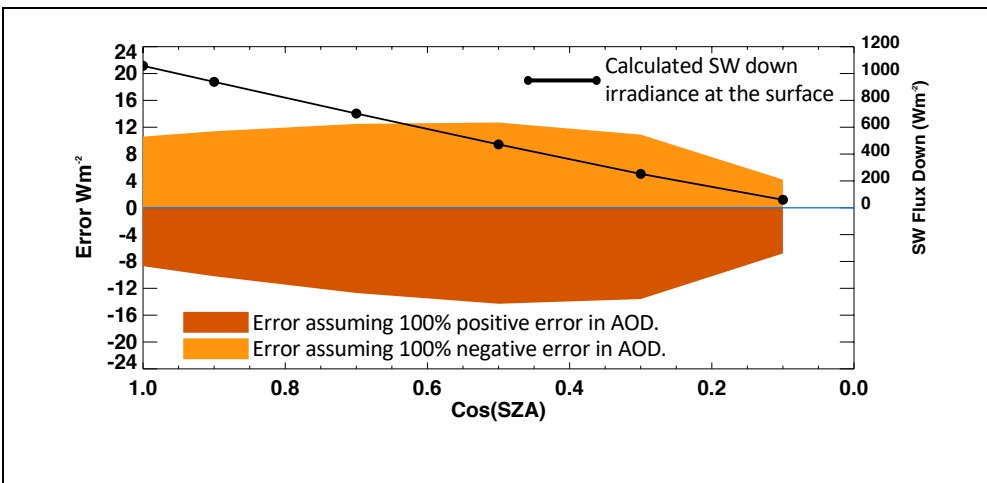

**Figure 10.** Estimated downward SW flux at the surface from the Langley Fu & Liou model along with estimated error in surface SW down +/-based on 100% error in AOD as a function of cos(SZA).



### 3.1 Effect of AOD differences on surface irradiances

**Figure 10** shows a summary of a series of radiative transfer calculations using the
"On-Line Langley Fu & Liou radiative transfer code:
https:// cloudsgate2.larc.nasa.gov/cgi-bin/fuliou/runfl.cgi
with an open shrub spectral albedo (broadband albedo of 0.14 at $\mu_0$=1.0), "continental"
aerosol, and no clouds. Values on the solid black line are the calculated surface
downward irradiances with an AOD of 0.09 at six different solar zenith angles.
Calculations were then done for AOD's of 0.0 and 0.18, at the same solar zenith angles,
representing 100% error bounds of the mean AODs derived from AERONET (as seen in
Tables 2 and 3 for the Australia sites where the RMS is approximately equal to the
observed average of AOD). The orange and red shaded areas indicate potential bias of
the downward shortwave irradiance at a given solar zenith angle. Irradiance values scale



nearly linearly between these limits.  **Figure 8** shows the error remains nearly constant
until a $\mu_0$=0.5 where it begins to decrease as insolation decreases. However, due to longer
path lengths at large solar zenith angles, the percentage error actually increases.

**4.  Clear Sky Comparisons of SYN1deg and Surface Observed Irradiances**

We complete our analysis of the impact of the MATCH aerosols on computed

surface irradiances by comparing calculated hourly mean surface downward shortwave
irradiance from the Ed4.1 SYN1deg-Hour product to observations of downward
shortwave irradiances. In a 1°×1° grid box with an approximate size of 111 km ×111 km,
100% clear sky sampled over one hour as determined by MODIS or geostationary
satellites is relatively rare. None the less, by grouping sites based on general surface
conditions and analyzing 20 years of data sufficient samples are found. **Figure 11** shows
the sites as grouped by color including 15 land sites labeled "Mid-Latitude" (Green
triangles), 6 sites labeled "Desert" (Red), 6 sites labeled "Polar" (White) and 46 buoys
(Blue).  Surface observed SW irradiance from the land sites comes from the Baseline
Surface Radiation Network (Ohmura et al. 1998; Dreimel et al. 2018) and buoy data are
made available



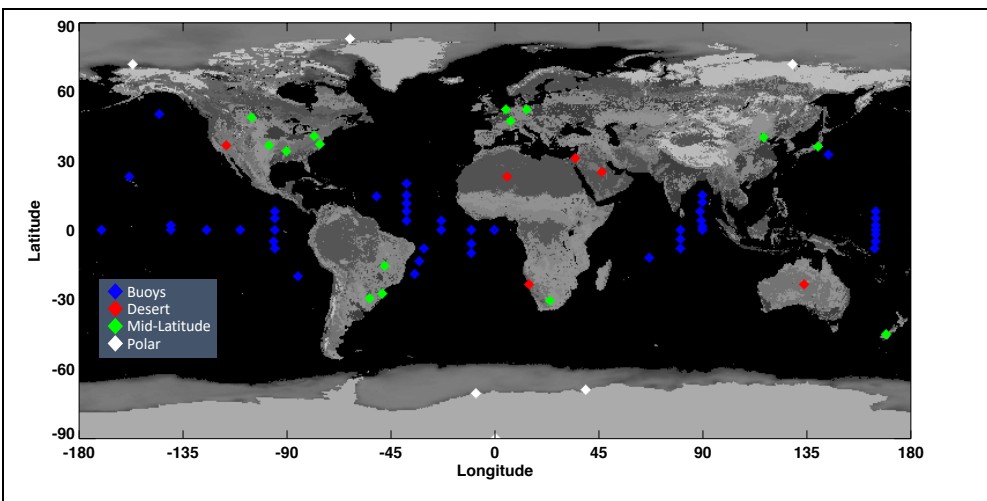

**Figure 11**. Location of surface observations of downwelling shortwave irradiance used to compare the SYN1deg Ed4.1 calculations to observations for all available hours (from Mar 2000 through Dec 2019) where the SYN1deg cloud analysis determines the hour to be 100% clear sky.


from the Pacific Marine Environmental Lab (PMEL) (McPhaden et al. 2002, 2009) and
the Woods Hole Oceanographic Institute (WHOI) (Colbo and Weller, 2009). A complete
listing is given in Appendix A.

**Figure 12** shows hourly comparisons of computed clear-sky downward

shortwave irradiance compared to observations for the four groups of sites shown in **Fig.**
**11**. In general, the calculated irradiance is larger than observed downward shortwave
irradiance. There we find that in every grouping, SYN1deg calculations tend to be too
transmissive, overestimating the surface downwelling SW irradiance by between 4 Wm$^{-2}$
(polar sites) and 16 Wm$^{-2}$ (ocean buoys) with mid-latitude and desert sites each
overestimating DSF by ~10 Wm$^{-2}$. It's notable that the smallest overestimation is in polar
regions where column AOD would be the smallest. This points to the possibility that
MATCH is weighted too far towards scattering aerosols and too few absorbing aerosols.

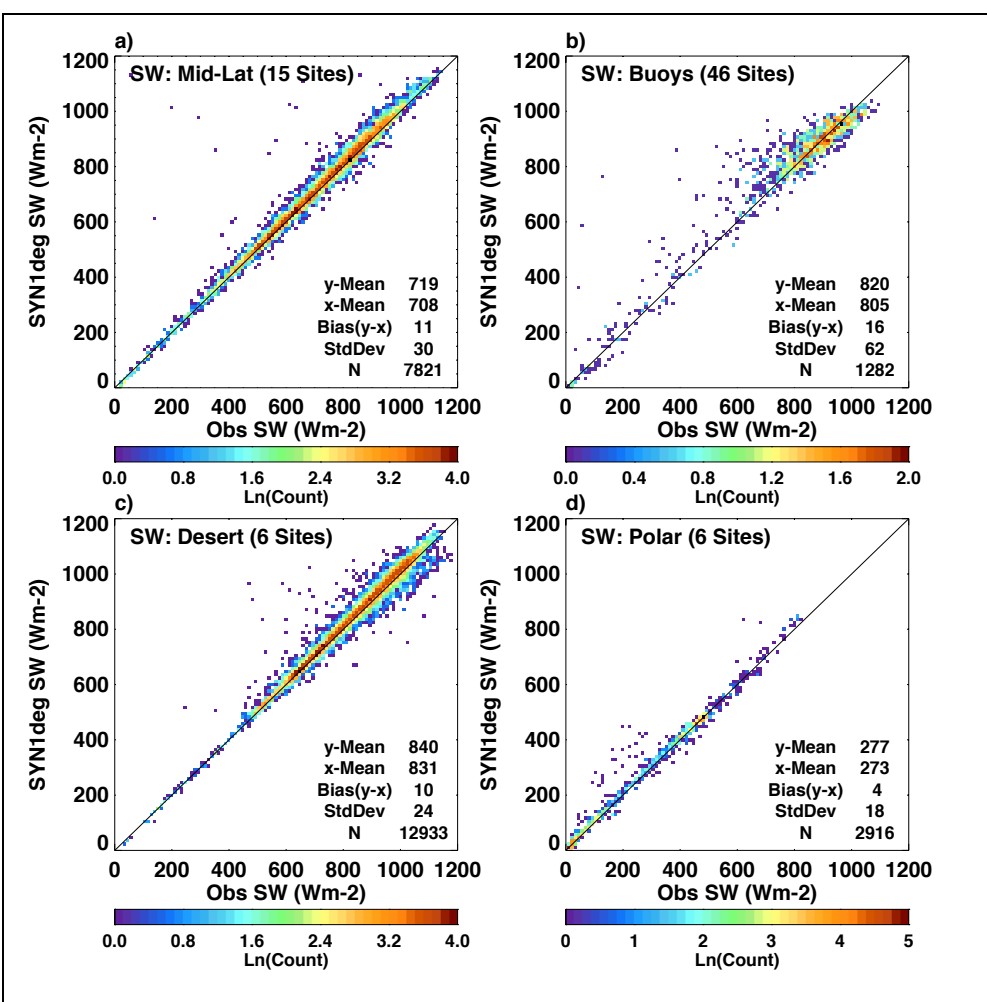

**Figure 12** Comparisons of SW downward irradiance at the surface from the SYN1deg Ed4.1 calculations (y-axis for all plots) and BSRN and buoy surface sites (x-axis all plots). Data are from Mar 2000 through Feb 2020 and only include hours when a 1x1 grid box is 100% clear-sky according to SYN1deg cloud fraction.


Clear-sky scenes used for **Fig. 12** are those identified by MODIS and

geostationary satellites over the 1°×1° grid where the ground site located. That is, when
the satellites did not detect clouds over the one-hour period within the grid box, we
compared the computed and observed hourly mean downward shortwave irradiances.
Clouds might have been present within the field-of-view of the ground-based



pyranometer but not within the field of view of sun-photometer. This would increase the
observed downward shortwave irradiance, hence the modeled irradiance would be
smaller. To verify, we used the ground-based cloud screening algorithm developed by
Long and Ackerman (Long et al. 2006) to further screen clouds. For the land groupings,
plots showing the difference between calculation minus observation as a function of
observation, utilizing both the satellite and surface based observed cloud fraction to 0.0,
are shown in **Fig. 13**. As shown in the plot, mean bias did not change significantly.
However, the RMS in both the Mid-Latitude and Desert sites was reduced by half due to
the more stringent cloud screening (**Fig. 13**).

**Figure 13**: Difference of computed and observed clear-sky downward shortwave irradiance at the surface as a function of observed surface irradiances. Each data point is hourly mean irradiance. Clear-sky is identified by MODIS and ground based observations by the Long-Ackerman algorithm. Top, middle and bottom plots are for midlatitude, desert, and polar groups shown in Figure 11.





CERES instruments observe TOA irradiances, which can be used to assess the
bias in computed irradiance. Global annual mean clear-sky TOA irradiances derived from
CERES observation averaged over 20 years from March 2000 through February 2020 are
53 Wm$^{-2}$ for reflected shortwave irradiance and 268 Wm$^{-2}$ for emitted longwave
irradiance. Corresponding computed reflected shortwave flux is 51 Wm$^{-2}$ and emitted
longwave flux is 267 Wm$^{-2}$. Insight into the surface irradiance errors may be gained by
considering how surface irradiance is modified via the tuning algorithm to match TOA
irradiance in the CERES EBAF-surface product (Kato et al. 2018). To match the
computed shortwave and longwave fluxes, AOD is increased from 0.136 to 0.156 (global
annual mean values) and precipitable water is decreased from 2.29 cm to 2.22 cm (global
annual mean values). These adjustments change the downward shortwave irradiance from
244 Wm$^{-2}$ to 243 Wm$^{-2}$.
To analyze how the EBAF tuning process changes surface irradiance, AOD and
precipitable water, we computed the mean change separated by surface group shown in
**Fig 12**. Generally, AOD increases and precipitable water decreases to increase reflected
shortwave flux, which in turn decreases surface downward shortwave irradiance over
these regions (**Table 3**). For the midlatitude group, on average, AOD is increased by
0.02, precipitable water is decreased by 0.06 cm, and surface albedo is increased by 0.03.
These adjustments reduce the diurnally averaged downward shortwave irradiance at the
surface by 2 Wm$^{-2}$. We do not have exact matches of BSRN and AERONET surface sites
but Tables 2 and 3 show MATCH AODs have either no bias (north America and China
and south East Asia) or slightly negatively biased by 0.01 (Europe). Therefore, increasing
MATCH AODs by 0.02 on average for the mid-latitude group seems justifiable.





However, decreasing 2 Wm$^{-2}$ for the diurnally averaged downward shortwave is smaller
than the 12 Wm$^{-2}$ bias shown in the top right plot of **Fig. 10**, although instantaneous
irradiances are used for **Fig. 10**. In addition, decreasing AODs for the desert group by
0.02 contradicts the positive bias (0.07) for the North Africa group shown in Table 2
under clear-sky conditions.

The adjustment made to match TOA shortwave irradiance, in the EBAF product,

is within the uncertainty of MODIS-derived AOD of ±0.05 over land and ±0.03 over
ocean (Remer et al. 2008; Levy et al. 2010, 2013). However, these are an expected error
of instantaneous AOD retrieval derived from the comparison of AODs with AERONET.
Therefore, the bias averaged over ground sites and many years is expected to be much
smaller. Although, the 0.03 AOD adjustment over ocean might be the upper limit of the
uncertainty of MODIS AODs over ocean, 16 Wm$^{-2}$ bias in the instantaneous downward
shortwave irradiance seems to be larger than the reduction by 2 Wm$^{-2}$ in the diurnally
averaged downward shortwave irradiance.

While we cannot identify the cause of the discrepancy between AOD comparison

and downward shortwave irradiance comparison with surface observations, potential
issues are following. 1) Aerosol type and optical properties used in irradiance
computations, and 2) bias in downward shortwave irradiance measured by pyranometer,
especially diffuse irradiance at smaller solar zenith angles. Because of the temperature
gradient within pyranometer, the downward shortwave irradiance measured by a
pyranometer tends to be biased low under clear-sky condition (Haeffelin et al. 2001).
Note that a study by Ham et al. (2020) indicates that the bias of diurnally averaged



surface downward shortwave irradiance computed by a four-stream model should be
smaller than 1%.

**Table 4**: Radiative flux, aerosol optical depth (AOD), precipitable water, and surface
albedo change to match observed top-of-atmosphere radiative fluxes

|  | | Changes: adjusted - unadjusted | | | | |
|---|---|---|---|---|---|---|
|  | Observed TOA upward shortwave irradiance (Wm⁻²) | Clear-sky TOA upward shortwave irradiance (Wm⁻²) | Clear-sky surface downward shortwave irradiance (Wm⁻²) | Clear-sky AOD | Clear-sky precipitable water (cm) | Clear-sky surface albedo |
| Mid-latitude | 63.3 | 3.9 | -2.0 | 0.02 | -0.06 | 0.03 |
| Desert | 92.3 | 3.4 | -1.7 | 0.02 | -0.04 | 0.01 |
| Polar | 86.5 | 8.2 | -0.2 | 0.01 | -0.03 | 0.10 |
| Buoys | 42.0 | 1.6 | -2.0 | 0.03 | -0.12 | 0.00 |


## 5. Conclusions

We evaluated MATCH aerosol optical depth used to produce the CERES

SYN1deg product. Aerosol optical depths derived from Terra and Aqua by the dark target
and deep blue algorithms were merged to produce daily gridded AODs. Daily gridded
AODs were used for assimilation by MATCH at local solar noon.  As a consequence,
monthly mean AODs under clear-sky conditions identified by MODIS closely agree with
those derived from MODIS, although MATCH uses climatological aerosol sources.
Because AODs are not screened by QAC, MATCH AODs are larger over convective
regions (e.g. Amazon, central Africa, and south east Asia) for both clear-sky and all-sky
conditions.

MATCH AODs under all-sky conditions are larger than those under clear-sky

conditions. Time series of AERONET AODs indicate that AODs generally increase with





cloud fraction, which is consistent with, primarily, water uptake by hygroscopic aerosols
(Varnai et al, 2017). In addition, surface observations at the ARM SGP site suggest that a
larger AODs and larger precipitable water under all-sky conditions than those under
clear-sky conditions. Aerosol optical depth biases from AERONET AODs are
comparable to biases of MERRA2 AOD biases from AERONET AODs for both all-sky
and clear-sky conditions. However, MERRA2, which uses AERONET AODs to train the
algorithm, has better temporal correlation with AERONET AODs than MATCH AODs.

Once MATCH AODs are used for surface irradiance computations, downward

shortwave irradiances are positively biased by 1% to 2% compared to those observed at
surface sites. Top-of-atmosphere reflected clear-sky shortwave irradiances are negatively
biased compared with those derived from CERES observations. Increasing AODs by
~0.02, and surface albedos by 0.03, and decreasing precipitable water by 0.06 cm over
mid-latitude surface sites makes computed reflected TOA irradiances agree with those
derived from CERES. These adjustments reduce downward shortwave irradiances at the
surface by 2 Wm$^{-2}$. Decreasing MATCH AODs for the desert group is needed to match
computed reflected shortwave irradiances at TOA with those derived from CERES.
However, decreasing MATCH AODs is not consistent with generally larger MATCH
AODs compared with AERONET.

Although optical properties of aerosols (i.e. aerosol type) play a minor role in

computing shortwave irradiance, changing aerosol type can alter the downward
shortwave irradiance in the right direction. We did not investigate the error in aerosol
type in this study. Aerosol types used in irradiance computations rely on those modeled
by MATCH. Biases in the fraction of each aerosol type and their optical properties can



change TOA upward and surface downward shortwave irradiances without altering total
AOD. Evaluation of aerosol type is left for the future study.

**Acknowledgments**

This work was funded by the NASA CERES project. The products and the

validation could not have been accomplished without the help of the CERES TISA team.
These data were obtained from the NASA Langley Research Center EOSDIS Distributed
Active Archive Center. We also wish to acknowledge the hard work put in by the many
dedicated scientists maintaining surface instrumentation in many diverse climates to
obtain high quality observations of downwelling shortwave and longwave surface flux.
Those groups are noted in Appendix A.

**Appendix A. Surface Observation Sites Used for Validation**

A great deal of data used in this study was collected by dedicated site scientists

measuring critical climate variables around the world. The tables included in this
appendix outline the sites, in situ measurements taken and their locations and dates of
available data. Table A1 lists the locations of the AERONET sites, our source for
observed aerosol optical depth which can be found on-line at:
https://aeronet.gsfc.nasa.gov/new_web/index.html.











| Table A1. AERONET Observation Sites | | | |
|---|---|---|---|
| Region | Site | Location | Available Months |
| North Africa (6 Sites) | Saada, Morocco | 31.6N, 8.2W | 2004/07 - 2019/04 |
| | Ouarzazate, Morocco | 30.9N, 6.9W | 2012/02 - 2015/06 |
| | Tenerife Isl., Spain | 28.3N, 16.5W | 2004/07 - 2019/04 |
| | Dhaka, Morocco | 23.7N, 15.9W | 2002/02 - 2005/11 |
| | Tamanrasset, Algeria | 22.8N, 8.2E | 2004/07 - 2019/04 |
| | Cape Verde Island | 16.7N, 22.9W | 2000/03 - 2018/12 |
| Central Africa (5 Sites) | Ilorin, Nigeria | 8.5N, 4.7E | 2000/03 - 2019/09 |
| | Koforidua, Ghana | 6.1N, 0.3W | 2012/12 - 2019/04 |
| | Lope, Gabon | 0.2S, 11.6E | 2014/04 - 2018/02 |
| | Mbita, Kenya | 0.4S, 34.2E | 2006/03 - 2017/17 |
| | Bujumbura, Burundi | 3.4S, 29.4E | 2013/12 - 2019/04 |
| China, Korea (8 Sites) | Xinglong, China | 40.4N, 117.6E | 2006/02 - 2014/11 |
| | Beijing, China | 39.9N, 116.4E | 2001/03 - 2019/03 |
| | Anymon Isl, S Korea | 36.5N, 126.3E | 2000/03 - 2019/11 |
| | Yonsei Univ, S Korea | 37.6N, 126.9E | 2011/03 - 2019/01 |
| | Cuiying Mt, China | 35.9N, 104.1E | 2006/07 - 2013/05 |
| | Nanjing, China | 32.2N, 118.7E | 2008/03 - 2010/04 |
| | Taihu, China | 31.4N, 120.2E | 2005/09 - 2016/08 |
| | XiangHe, China | 39.7N, 116.9E | 2001/03 - 2017/05 |
| India, SE Asia (8 Sites) | Gandhi College, India | 25.8N, 84.1E | 2006/04 - 2019/11 |
| | Luang Namtha, Laos | 20.9N, 101.4E | 2001/04 - 2019/02 |
| | Omkoi, Thailand | 17.8N, 98.4E | 2003/02 - 2018/03 |
| | Dhaka Univ, Bangledesh | 23.7N, 90.3E | 2012/06 - 2019/07 |
| | Bhola, Bangledesh | 22.2N, 90.7E | 2013/04 - 2019/04 |
| | Nghia Do, Vietnam | 21.0N, 105.8E | 2010/11 - 2019/09 |
| | Pune, India | 18.5N, 73.8E | 2004/10 - 2019/06 |
| | Hanimaadhoo, Maldives | 6.7N, 73.2E | 2004/11 - 2019/09 |






















| Table A1. AERONET Observation Sites (Continued) | | | |
|---|---|---|---|
| Region | Site | Location | Available Months |
| Brazil (7 Sites) | Petrolina, Brazil | 9.1S, 40.4W | 2004/07 - 2016/11 |
| | Abracos Hill, Brazil | 10.7S, 62.4W | 2000/03 - 2005/10 |
| | Alta Floresta, Brazil | 9.9S, 56.1W | 2000/05 - 2019/02 |
| | Belterra, Brazil | 2.6S, 55.0W | 2000/03 - 2005/04 |
| | Ji Parana SE, Brazil | 10.9S, 61.9W | 2006/01 - 2017/10 |
| | Manaus, Brazil | 2.9S, 60.0W | 2011/02 - 2019/05 |
| | Rio Branco, Brazil | 9.9S, 67.9W | 2000/07 - 2017/10 |
| Australia (6 Sites) | Jabiru, Australia | 12.6S, 132.9E | 2000/03 - 2019/09 |
| | Lake Argyle, Australia | 16.1S, 128.7E | 2001/10 - 2019/09 |
| | Canberra, Australia | 35.3S, 149.1E | 2003/01 - 2017/08 |
| | Birdsville, Australia | 25.9S, 139.3E | 2005/08 - 2018/06 |
| | Lucinda, Australia | 18.5S, 146.4E | 2009/10 - 2020/01 |
| | Lake Lefroy, Australia | 31.2S, 121.7E | 2012/06 - 2019/12 |
| North America (10 Sites) | Brats Lake, Canada | 50.2N, 104.7W | 2000/03 - 2013/02 |
| | Sioux Falls, SD | 43.7N, 96.6W | 2001/06 - 2017/10 |
| | Ames, IA | 42.0N, 93.8W | 2004/05 - 2019/03 |
| | Boulder Tower | 40.0N, 105W | 2001/05 - 2016/07 |
| | Bondville, IL | 40.0N, 88.4W | 2000/03 - 2017/10 |
| | Brookhaven, NY. | 40.8N, 72.9W | 2002/09 - 2020/01 |
| | Wallops Island, VA | 37.9N, 75.5W | 2003/03 - 2020/03 |
| | ARM/SGP E13 | 36.6N, 97.5W | 2000/03 - 2018/05 |
| | Chesapeake Light Tower | 36.9N, 75.7W | 2000/03 - 2016/01 |
| | Table Mountain, CO | 40.1N, 105.2W | 2008/11 - 2017/12 |
| Europe (10 Sites) | Cabauw, Netherlands | 51.9N, 4.9E | 2003/04 - 2017/11 |
| | Palaiseau, France | 48.7N, 2.2E | 2000/03 - 2020/10 |
| | Torevere, Estonia | 58.2N, 26.5E | 2002/06 - 2019/07 |
| | Kishinev, Moldova | 47.0N, 28.8E | 2000/03 - 2018/11 |
| | Belsk, Poland | 51.8N, 20.8E | 2004/01 - 2016/08 |
| | Kyiv, Ukraine | 50.3N, 30.5E | 2007/04 - 2018/12 |
| | Hamburg, Germany | 53.5N, 9.9E | 2000/06 - 2018/06 |
| | Munich Univ, Germany | 48.1N, 11.6E | 2001/11 - 2019/05 |
| | Thessaloniki, Greece | 40.6N, 22.1E | 2003/06 - 2020/03 |
| | Bucharest, Hungary | 44.3N, 26.0E | 2000/10 - 2019/03 |


Sources of surface observed downwelling irradiance are outlined in Tables
A2 (land) and A3 (buoys). For land we utilize data from the Baseline Surface
Radiation Network (BSRN) (Dreimel et al, 2018; Ohmura et al. (1998)), the US Dept.
of Energy's Atmospheric Radiation Measurement (ARM) program and NOAA's
SURFRAD network available from NOAA's Air Resources Laboratory/Surface
Radiation Research Branch., Augustine et al. (2000). Buoy observations come from





two sources through four separate projects. The Upper Ocean Processes group at
Woods Hole Oceanographic Institute have maintained the Stratus, North Tropical
Atlantic Site (NTAS) and Hawaii Ocean Time Series (HOTS) buoys for more than a
decade providing valuable time series of radiation observations in climatically
important regions of the ocean.  These data can be retrieved from:
http://uop.whoi.edu/index.html. We would also like to acknowledge the Project
Office of NOAA's Pacific Marine Environmental Labs (PMEL) where three groups of
buoy data were downloaded: In the Pacific, the Tropical Atmosphere
Ocean/Triangle Trans-Ocean Buoy Network (TAO/TRITON) (McPhaden, 2002) data,
from the tropical Atlantic Ocean, the Prediction and Research Moored Array in the
Tropical Atlantic (PIRATA) (Servain et al. 1998), and the Research Moored Array for
African - Asian - Australian Monsoon Analysis and Prediction (RAMA) (McPhaden et
al., 2009) in the Indian Ocean. Also downloaded from PMEL are the long-term buoy
observations PAPA and Kuroshio Current observatory sites.













### Table A2. Surface Irradiance Validation Sites (Land)

| Region | Site | Location | Source |
|---|---|---|---|
| Mid-Latitude (15 Sites) | Lindenberg, Germany | 52.2N, 14.1E | BSRN |
| | Cabauw, Netherlands | 51.9N, 4.9E | BSRN |
| | Fort Peck, MT | 48.3N, 105.1W | BSRN |
| | Payerne, Switzerland | 46.8N, 6.9E | BSRN |
| | Penn State, PA | 40.7N, 77.9W | SURFRAD |
| | Beijing, China | 39.9N, 116.3E | BSRN |
| | E13, Lamont, OK | 36.6N, 97.5W | ARM |
| | Ches Light Tower, USA | 36.9N, 75.7W | BSRN |
| | Tateno, Japan | 36.1N, 140.1E | BSRN |
| | Goodwin Creek, MS | 34.2N, 89.9W | SURFRAD |
| | De Aar, South Africa | 30.6S, 24.0E | BSRN |
| | Lauder, New Zealand | 45.0S, 169.7E | BSRN |
| | Florianapolis, Brazil | 27.5S, 48.5W | BSRN |
| | Brasilia, Brazil | 15.6S, 47.7W | BSRN |
| | Sao Martinho da Serra, Brazil | 29.4S, 53.8W | BSRN |
| Desert (6 Sites) | Sede Boqer, Israel | 30.8N, 34.7E | BSRN |
| | Saudi Solar Village | 24.9N, 46.4E | BSRN |
| | Tamanrasset, Algeria | 22.8N, 5.5E | BSRN |
| | Desert Rock, NV | 36.6N, 116.1W | SURFRAD |
| | Alice Springs, Australia | 23.7S, 133.8E | BSRN |
| | Gobabeb, Namibia | 23.5S, 15.0E | BSRN |
| Polar (6 Sites) | Alert, Canada | 82.5N, 62.4W | BSRN |
| | Tiksi, Russia | 71.6N, 128.9E | BSRN |
| | Barrow, Alaska | 71.3N, 156.7W | BSRN |
| | Syowa, Antarctica | 69.0S, 39.5E | BSRN |
| | South Pole, Antarctica | 90.0S, 0.5E | BSRN |
| | G. von Neumayer, Antarctica | -70.6S, 8.3W | BSRN |

BSRN: Baseline Surface Radiation Network, http://bsrn.awi.de/
SURFRAD: NOAA- SURFace RADiation Program, http://www.esrl.noaa.gov/gmd/grad/surfrad/
ARM: US Dept of Energy, Atmospheric Radiation Measurement Program, http://www.arm.gov/

### Table A3. Surface Observation Sites for Ocean Buoy Locations

| Program Name | Data Source | Locations |
|---|---|---|
| Upper Ocean Processes Group (UOP) 3 Buoys | Woods Hole Oceanographic Institute | Stratus Buoy -20.2N, 85.0W |
| | | North Tropical Atlantic Buoy 14.5N, 51.0W |
| | | Hawaii Ocean Time Series Buoy 22.5N, 158W |
| PIRATA Buoys 14 Buoys | Pacific Marine Environmental Laboratory (PMEL) | East Atlantic Ocean |
| RAMA Buoys 10 Buoys | PMEL | Tropical Indian Ocean |
| TAO Array Buoys 17 Buoys | PMEL | E & W Tropical Pacific Ocean |
| Kuroshio Extension Observatory Buoy | PMEL | NW Pacific, 32.4N, 144.6E |
| PAPA Sub-Arctic Ocean Buoy | PMEL | NE Pacific, 50.1N, 144.8W |

UOP: http://uop.whoi.edu/projects/projects.htm
PMEL: http://www.pmel.noaa.gov/tao/data_deliv/deliv.html




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
