# Peer review of "Evaluation of aerosol optical depths and clear-sky radiative fluxes of the"

_Atmospheric Chemistry and Physics, 2021_

## Referee Comment (RC1)

Review of paper:

**Evaluation of aerosol optical depths and clear-sky radiative fluxes of the CRERES Edition 4.1 SYN1deg data product by D.Fillmore et al.**

**Positives**

- Overdue assessment of Match aerosol properties used in the CERES data product
- Comparisons between MATCH and MERRA
- Using AERONET and BSRN references
- Efforts to match scales and conditions in data-comparisons

**Concerns**

- focus on AOD (also modulated by prescribed water uptake), while rad.flux disagreements are also effected by aerosol absorption (varies regionally/seasonally) and aerosol size.
- focus only on shortwave radiative (clear-sky) closure, while the added use of the longwave radiative (clear-sky) closure would provide extra constraints.
- Limiting AERONET data analysis to AOD, while detailing and complementing data (AODf, AODc, AAOD and even water vapor) are available.
- Ignoring a focus on (wildfire and dust) seasons, where aerosol signals are stronger.

**General comments:**

The paper investigates the quality of the aerosol representations of the Match model in radiative transfer applications for the surface (and atmospheric) radiation flux product. Considering that the surface (and atmospheric) energy budget are important for surface processes (and atmospheric dynamics) this contributions is overdue. First AOD data are compared to MODIS noon-time (retrievals) and to Merra (assimilations), then spatially limited (sparse, land-based) comparisons to AERONET data are analyzed and finally the calculated clear-sky downward radiative fluxes are compared to (also) spatially limited BRSN data. Finally, also necessary corrections, not just limited to aerosol properties are discussion so that closure with the (stronger to observed tied) TOA fluxes are achieved.

There are apparent limitations, that cannot be changed, such as (1) the climatological emissions (without the ability to cover specific dust or wildfire events ... other than general noon-time (total) AOD adjustments with MODIS retrieval data) and (2) assumed aerosol types (refractive indices, size, water uptake).

Many of the offered comparisons are interesting. For instance the separation between less cloudy and cloudy events, the relatively strong (stronger than in Merra) aerosol water uptake is revealed. Also, the separation of the analysis to the reference (AERONET, BSRN) into region for the focus on the representation biases of specific aerosol is commendable but in the end the final conclusions leave more questions than answers.

In order to get closure with TOA CERES clear-sky data, the full potential of aerosol properties is not investigated by rather changes to water vapor and solar surface albedo are 'invited'. This is disappointing from an aerosol property perspective (AERONET offers a split into fine and coarse mode AOD contributions, AAOD absorption data for high aerosol load cases on which I would focus first, and even water vapor column data, for LW closure) and from a radiative flux (closure) perspective (as SW data are also accompanied by LW data).

I know that the paper has already some volume but the last sentence of the conclusion is not satisfying at all.

On another note, I would also look to compare the (clear-sky) surface (and also TOA) radiation budgets of CERES (with the Match data) to those in Langley SRB product. Maybe in this context even aerosol optical and radiative (also component) properties of MACv2 (derived from optics) could be compared.

The MACv2 (1x1, mon) aerosol climatology is available via anonymous ftp … and even single scattering properties for your 31 (17 +14) band spectral resolution are available ftp-projects.zmaw.de/aerocom/climatology/MACv2\_2018 all subdirectories are useful … but those 4 most useful for you /documents - documentation in papers /550nm – 550nm properties for the reference year /spectral/ssp\_31bands - your spectral resolution for the 2010 reference year /spectral/ssp\_31bands - aeback … are pre-sel aero components with assumed opt properties) /spectral/ssp\_31bands/by\_years - your spectral res. for indiv.years (only anthro. AODf changes)

(for access: use filezilla [FIREFOX does not work anymore] or use lftp from a linux machine)

**Minor comments:**

Line 33 "AOD are greater at all-sky conditions" not necessarily in modeling ... "supported by AERONET" how? as AERONET only samples at cloud-free view.

Line 35 Merra like Match are also using MODIS data ... so we compare to model applications of satellite retrievals (which could be incorrect and biased) but not against the truth.

Line 46 Differences can be attributed not just to amount (AOD) but also to aerosol absorption and aerosol size (SSA). For larger differences in case of dust aerosol it would be also great to look for consistency with broadband longwave fluxes. (This however requires not only good data on dust size and dust IR absorption but also accurate data for water vapor profiles and dust altitude).

Line 140 are the needed ancillary data (10m winds, climatology oxidants fields and emission inventories .. what databases?) for each specific year (since 2000) applied ... or are these based only a general climatology?

Line 149-158 the assumptions for the aerosol component optical properties are essential, thus at least component reff and RFimag at 550nm (and for dust components also the RFI at 10um) and resulting SSA values at 550nm for all components should be listed. I also assume a component external mixing ... correct?

Line 183 By assuming that AOD differences drive the changes in aerosol mass, it is likely postulated that the model assumed mixtures and mass extinctions apply, which may not be the case especially if AOD changes are associated with a singular aerosol component. In that sense it would be better to assimilate at least fine-mode and coarse mode properties separately as their properties are quite different. Another limitation is that the fine and coarse mode assumption for absorption and size in MODIS retrievals likely differ from those assumed in the Match model, so that the AOD comparison are biased.

Line 197 it is interesting that Merra considers more input than just MODIS AOD ... but I wonder in what way other day will and can override MODIS AOD data... thus I think the Merra data cannot really serve as reference. (In that context also the aerosol component treatment is likely different). Figure 2 line 205 the multi-annual differences between Merra and Match are large and even often with different signs in some regions (and more than 0.3 is very large!)

Line 215 could not the daily MODIS data be subsampled for improved Match model comparisons (although MODIS may be still biased). Then it also would be interesting to compare fine-mode AOD and coarse mode AOD. On the other hand more than 15 years of data twice a data averaged (via monthly) into multi-annual global averages should provide sufficient statistics.

Line 220 the number of MODIS AOD –compared to other AOD data- will dominate so differences in AOD input between Merra and Match should not matter that much

Line 224 AOD are generally smaller over ocean than over land (expect for outflow regions) so that AOD differences should be smaller. Generally, oceanic assimilated oceanic AOD differences are smaller although Merra has significantly larger (MODIS-like) AOD over the southern oceans.

Line 225 the sharp land/sea contrast off northern Africa in differences (likely a model feature) are concerning, as well Merra's much lower values over SE and E Asia and Match's larger values over E Asia, E. Europe/N.Asia and over the Andes mountains stand out. I would investigate in what way a too strong absorbing aerosol type in the model would cause an AOD overestimate (e.g. in Match) or a too waekly absorbing aerosol type in the model would cause an AOD underestimate (e.g. in Merra) – assuming that the retrieval absorption assumption are correct. (It also would not hurt to compare SSA maps between Merra and Match to better understand the large AOD differences in the models in some regions).

Line 233 the log/log plot choice hides the (MODIS high) bias at lower AOD values. Nonetheless the scatter (consider that these are assimilation) are considerable. Overall, the assimilated global annual AOD values are lower at 0.142 in Merra and 0.160 in Match. It is known that MODIS is biased high (mainly at lower AOD values) so the relatively large global average in Match surprises. It would be nice to compare by AOD-bin and by region/seasons at least for ideas on how to find clues for better comparisons ... so Figure 4 offers some insights but many regional differences are so large that they need some explanation.

Figure 4 Line 261 why just 1 month (jan 2020) and not 15+years? It is interesting that AOD increase from clear-sky to all-sky in Match apparently only happens pollution regions over E. Asia US and Europe, which begs the question if assumed ambient rel. humidity aerosol water uptake in Match at higher rel.hum is too strong (also clouds with 100% hum. only occupy specific layers). And with clouds in the neighborhood there is not only the issue with aerosol size swelling but also the issue with wet removal. Thus, global models are highly diverse if AOD under all-sky are actually larger or smaller with respect to the clear-sky value. The large difference over biomass regions suggest that aerosol (fine-mode) absorption is much stronger in Match. Thus I would compare fine-mode SSA and water uptake between both models.

Line 268 ... and aerosol absorption (and dust size)

Figure 5 line 291, Bejing is an urban difficult site with a lot of near surface pollution during winter. Both models suggest larger AOD with clouds, whereby the increase is much stronger in Match than Merra. Otherwise the AERONET values are larger, most likely due to nitrate aerosol, which is missing in Match and possibly also Merra.

Line 304 the cases with less than 1% (in 1deg regions) are probably too few for useful statistics. (I would split all cases in ca equal counts with more and less clouds). I am bit surprised than with the 1% threshold relative many cases are found ... so do I misinterpret here? Also not that in the comparisons many AERONET sites are more local (urban) and not regionally representative.

Line 325 ... would this make you want to reduce the water uptake in Match?

Table 2 Line 341 I wonder about the many matches for Australia, the small AOD values over Brazil with the biomass burning, the relatively large AOD values over central Africa (although the sites are spread. and the small values for N.Africa associated with dust. I strongly recommend to use AERONET as reference not just for AOD, but also for the split in AODf and AODc and also for AAOD data (at larger AERONET AOD cases AOD550 > 0.25). Given the large differences between the models in some regions (in Fig 4) the bias differences among the models are relatively small ... why? Line 440 Figure 10 (not 8)

Line 469 also dust size is an issue (as larger dust for the same mid-vis AOD has lower SSA and also larger spectral near-IR AOD)

Line 511 I do not follow the argument of a likely lower dust AOD ... as such a tendency seems to go the right direction ... given the 0.07 bias in Table 2...??

Line 560 I wonder about the surface albedo treatment (at 14% solar broadband values are mentioned and now a significant 3% increase was mentioned). I hope the Kato model uses the spectral dependency for local soil conditions from MODIS. I would tinker not just with AOD but also with size and absorption before starting to change the surface albedo.

Line 571 ... exactly ... and NO it should be part of this study.

---

## Author Comment (AC1)

Review of paper:

**Evaluation of aerosol optical depths and clear-sky radiative fluxes of the CRERES Edition 4.1 SYN1deg data product          *by D.Fillmore et al.**

**Positives**
- Overdue assessment of Match aerosol properties used in the CERES data product
- Comparisons between MATCH and MERRA
- Using AERONET and BSRN references
- Efforts to match scales and conditions in data-comparisons

**Concerns**
- focus on AOD (also modulated by prescribed water uptake), while rad.flux disagreements are also effected by aerosol absorption (varies regionally/seasonally) and aerosol size.
- focus only on shortwave radiative (clear-sky) closure, while the added use of the longwave radiative (clear-sky) closure would provide extra constraints.
- Limiting AERONET data analysis to AOD, while detailing and complementing data (AODf, AODc, AAOD and even water vapor) are available.
- Ignoring a focus on (wildfire and dust) seasons, where aerosol signals are stronger.

*The authors would like to thank this reviewer for the careful reading and keen interest in the work presented.  Throughout the review interesting comments and valuable suggestions are given and many are acted upon. However, we would like to address some of these major concerns, which echo throughout the review, at this time. The paper has two primary foci. The first is to describe the MATCH chemical transport model and its interface with the radiative transfer calculation in the CERES SYN1deg product, and the second to validate the resultant global AODs used in the radiative transfer calculations available from the CERES SYN1deg data product. In summary of the paper's introduction, we state: "In this paper, we evaluate* aerosol optical depth used for irradiance computations *in the CERES project and analyze how the error propagates to clear-sky surface irradiances." And though the reviewer has many thought-provoking ideas for further analyses we cannot address many at this point as it would require a rewriting of significant portions of code and re-running or the operational process that produces the SYN1deg product. For example, the reviewer suggests several times the separation of our results into more detailed analyses of aerosol species/properties with respect to their effect on radiative transfer calculations. However, the MATCH model does not resolve the aerosol size. Therefore, assimilation of fine and coarse modes separately is not possible though we do attempt such a comparison using our dust AOD components. Results are shown in our response to minor comment 18 below and are shown to be inconclusive.  (We have ordered the reviewer's*

*comments and responses are in italics. Figure and line numbers in our response refer to the original paper's.)*

**General comments:**

A. The paper investigates the quality of the aerosol representations of the Match model in radiative transfer applications for the surface (and atmospheric) radiation flux product. Considering that the surface (and atmospheric) energy budget are important for surface processes (and atmospheric dynamics) this contributions is overdue. First AOD data are compared to MODIS noon-time (retrievals) and to Merra (assimilations), then spatially limited (sparse, land-based) comparisons to AERONET data are analyzed and finally the calculated clear-sky downward radiative fluxes are compared to (also) spatially limited BRSN data. Finally, also necessary corrections, not just limited to aerosol properties are discussion so that closure with the (stronger to observed tied) TOA fluxes are achieved.

There are apparent limitations, that cannot be changed, such as (1) the climatological emissions (without the ability to cover specific dust or wildfire events … other than general noon-time (total) AOD adjustments with MODIS retrieval data) and (2) assumed aerosol types (refractive indices, size, water uptake).

Many of the offered comparisons are interesting. For instance the separation between less cloudy and cloudy events, the relatively strong (stronger than in Merra) aerosol water uptake is revealed. Also, the separation of the analysis to the reference (AERONET, BSRN) into region for the focus on the representation biases of specific aerosol is commendable but in the end the final conclusions leave more questions than answers.

In order to get closure with TOA CERES clear-sky data, the full potential of aerosol properties is not investigated by rather changes to water vapor and solar surface albedo are 'invited'. This is disappointing from an aerosol property perspective (AERONET offers a split into fine and coarse mode AOD contributions, AAOD absorption data for high aerosol load cases on which I would focus first, and even water vapor column data, for LW closure) and from a radiative flux (closure) perspective (as SW data are also accompanied by LW data).

I know that the paper has already some volume but the last sentence of the conclusion is not satisfying at all.

*Though the last several sentences in our conclusion may not be satisfying to the reviewer, they do state succinctly the current state of our ability to evaluate outputs from the MATCH model and impact on radiative transfer results found in the SYN1deg product. We use comparisons of clear sky shortwave irradiance calculations with observations as a tool to measure MATCH inputs, not as a study of radiative closure. Though obviously closely intertwined, a full discussion of radiative closure, or lack thereof, within the SYN1deg product, is beyond the scope of this paper. We address the use of fine and coarse mode observations from AERONET in our response to comment 19. from the reviewer.*

B. On another note, I would also look to compare the (clear-sky) surface (and also TOA) radiation budgets of CERES (with the Match data) to those in Langley SRB product. Maybe in this context even aerosol optical and radiative (also component) properties of MACv2 (derived from optics) could be compared.

C. The MACv2 (1x1, mon) aerosol climatology is available via anonymous ftp … and even single scattering properties for your 31 (17 +14) band spectral resolution are available
ftp-projects.zmaw.de/aerocom/climatology/MACv2_2018 all
subdirectories are useful … but those 4 most useful for you
/documents  - documentation in papers
/550nm – 550nm properties for the reference year
/spectral/ssp_31bands  - your spectral resolution for the 2010 reference year
/spectral/ssp_31bands  - aeback … are pre-sel aero components with assumed opt properties)
/spectral/ssp_31bands/by_years  - your spectral res. for indiv.years (only anthro. AODf changes)

(for access: use filezilla [FIREFOX does not work anymore] or use lftp from a linux machine)

*Release 3 of the Langley SRB product used a simpler aerosol scheme based on continental and marine aerosol types whose initial values were drawn from modal values from MATCH results. SRB Release 4 uses the MAC v1. However MACv1 provides climatological mean aerosol optical thickness. Hence it misses large events such as smoke and volcanic ash. We do not think that a comparisons of climatological means add any additional value to the manuscript as we already include significant comparisons to AERONET observations at higher temporal resolution.*

**Minor comments:**

1. Line 33  "AOD are greater at all-sky conditions" not necessarily in modeling … "supported by AERONET"  how? as AERONET only samples at cloud-free view.

*Though AERONET inversions require their cloud masking algorithm to indicate 'clear', Figure 8b) of the manuscript shows AERONET observations taken under various cloud fractions as determined by upward looking surface shortwave radiometers via the Shortwave Flux Analysis (SWFA) algorithm (Long et al, 2006). The figure delineates a distinct difference in AOD observed when the SWFA indicates clear sky (SWFA=0.0) relative to when it does not. Likewise, our primary source of cloud fraction for this study is based on the percentage of clouds in a 1deg grid box in which an AERONET site is located. Figure 9 shows that as the satellite-based cloud fraction increases within a 1-degree grid box AERONET retrievals of AOD also increase. We elaborate on the relationship between AERONET observation, cloud fraction and water vapor in Section 3.*

2. Line 35 Merra like Match are also using MODIS data … so we compare to model applications of satellite retrievals (which could be incorrect and biased) but not against the truth.

*We produce a global product and so compare our results with those from MERRA to give a global picture. We state that MERRA assimilates MODIS AOD, as well as some AERONET and fire information, which implies the MERRA data is not entirely independent of our results. After MODIS assimilation MATCH remains slightly closer to MODIS AOD which we consider to be a good result. 'Truth', in so much as it can be defined, is reserved for comparison with AERONET which we discuss in detail and quantify in Tables 2 and 3 against both MATCH and MERRA results.*

3. Line 46 Differences can be attributed not just to amount (AOD) but also to aerosol absorption and aerosol size (SSA). For larger differences in case of dust aerosol it would be also great to look for consistency with broadband longwave fluxes. (This however requires not only good data on dust size and dust IR absorption but also accurate data for water vapor profiles and dust altitude).

*We note that a 20% decrease in column precipitable water commonly results in a decrease in downward longwave irradiance (DLF) at the surface of ~4%. On the other hand, keeping water vapor fixed and changing aerosol from 1 micron dust to water soluble sulfate aerosol reduces the DLF by ~0.8%. Given that we have found good agreement between the GEOS5.4.1 column precipitable water with micro-wave radiometer observations we decided to not extend the paper with analyses showing LW comparisons.*

*We understand that differences are attributable to more than just AOD, however the output we retained from MATCH and radiative transfer model runs did not include the combined aerosol properties. That is, the various constituents (and so their associated scattering/absorption properties) are weighted and averaged for each RTM run. We did not retain that information. This makes it almost impossible to 'unscramble the egg' of the aerosol properties that were used in the RTM calculations without modifying and re-running the production code which is not possible at this time.*

4. Line 140 are the needed ancillary data (10m winds, climatology oxidants fields and emission inventories .. what databases?) for each specific year (since 2000) applied … or are these based only a general climatology?

*In Line 114/115 we state that "MATCH is a transport model …driven by offline meteorological fields from …NCEP reanalysis." Table 1 (and it's description in lines 138-148) lists details of how each aerosol type is modeled and whether or not it is based on a climatology.*

5.  Line 149-158 the assumptions for the aerosol component optical properties are essential, thus at least component reff and RFimag at 550nm (and for dust components also the RFI at 10um) and resulting SSA values at 550nm for all components should be listed. I also assume a component external mixing … correct?

*For the aerosol types used in the running of the Langley Fu & Liou radiative transfer model we have added a plot (now Fig 1) of SSA and g at 550 nm. Hygroscopic aerosols are shown as a function of relative humidity.*

6.  Line 183 By assuming that AOD differences drive the changes in aerosol mass, it is likely postulated that the model assumed mixtures and mass extinctions apply, which may not be the case especially if AOD changes are associated with a singular aerosol component. In that sense it would be better to assimilate at least fine-mode and coarse mode properties separately as their properties are quite different. Another limitation is that the fine and coarse mode assumption for absorption and size in MODIS retrievals likely differ from those assumed in the Match model, so that the AOD comparison are biased.

*This is an interesting idea but outside the scope of this research.  The MATCH model does not resolve the aerosol size. Therefore, assimilation of fine and coarse modes separately is not possible.*

7.  Line 197 it is interesting that Merra considers more input than just MODIS AOD … but I wonder in what way other day will and can override MODIS AOD data… thus I think the Merra data cannot really serve as reference. (In that context also the aerosol component treatment is likely different). Figure 2 line 205 the multi-annual differences between Merra and Match are large and even often with different signs in some regions (and more than 0.3 is very large!)

*That MERRA assimilates more information than MODIS retrievals does not remove it as a source of comparison. In fact, it enhances the comparison by showing places/times (such as the recent spate of intense forest fires) where MATCH is dependent upon MODIS identifying the increase in AOD. It points to the need to update MATCH to be more responsive to high time resolution events and to better identify the source of such event. For example, though MATCH may assimilate the AOD from a smokey fire, after the AOD is viewed by MODIS and assimilated, it does not know the source of the aerosol and so misses the increase in black carbon from such an event.*

8.  Line 215  could not the daily MODIS data be subsampled for improved Match model comparisons (although MODIS may be still biased). Then it also would be interesting to

compare fine-mode AOD and coarse mode AOD. On the other hand more than 15 years of data twice a da(y) averaged (via monthly) into multi-annual global averages should provide sufficient statistics.

*To return to the model inputs and sample MATCH values at MODIS observation times constitutes a significant effort that, while it might be interesting, does not change our results and will not add significant content to the paper.*

9. Line 220 the number of MODIS AOD –compared to other AOD data- will dominate so differences in AOD input between Merra and Match should not matter that much

*This is true and we've added to this sentence " when and where these events might occur."*

10. Line 224 AOD are generally smaller over ocean than over land (expect for outflow regions) so that AOD differences should be smaller. Generally, oceanic assimilated oceanic AOD differences are smaller although Merra has significantly larger (MODIS-like) AOD over the southern oceans.

*It does appear that the MERRA product is slightly higher than MODIS along the storm tracks in the southern ocean.*

11. Line 225 the sharp land/sea contrast off northern Africa in differences (likely a model feature) are concerning, as well Merra's much lower values over SE and E Asia and Match's larger values over E Asia, E. Europe/N.Asia and over the Andes mountains stand out. I would investigate in what way a too strong absorbing aerosol type in the model would cause an AOD overestimate (e.g. in Match) or a too waekly absorbing aerosol type in the model would cause an AOD underestimate (e.g. in Merra) – assuming that the retrieval absorption assumption are correct. (It also would not hurt to compare SSA maps between Merra and Match to better understand the large AOD differences in the models in some regions).

*The purpose of Figure 2) is to describe, succinctly, differences between the MODIS AOD, which is assimilated by both MATCH and MERRA, to the AOD's output by these models after the assimilation process. We discuss in detail the tendency of MATCH to overestimate AOD over land due to the dominance of the influence of aerosol climatology under cloudy sky conditions throughout the discussion portion of the paper. While it is interesting, for instance over the equatorial rain forests to find differences of opposite sign, we can only speculate on why this might be so for MERRA.*

12. Line 233 the log/log plot choice hides the (MODIS high) bias at lower AOD values. Nonetheless the scatter (consider that these are assimilation) are considerable. Overall, the assimilated global annual AOD values are lower at 0.142 in Merra and 0.160 in Match. It is known that MODIS is biased high (mainly at lower AOD values) so the relatively large global

average in Match surprises. It would be nice to compare by AOD-bin and by region/seasons at least for ideas on how to find clues for better comparisons ... so Figure 4 offers some insights but many regional differences are so large that they need some explanation.

*Figure 3 is in essence, a scatter plot analog to Figure 2. It encompasses a large number of comparisons over a significant time span, 20 years. Hence the need for the log density plot. Figure 3 says that, compared with MODIS AOD, MERRA-2 AOD is lower by 0.032 and MATCH AOD is lower by 0.014. MODIS AOD is higher Scatter is large with RMS of each model approaching 75% of the mean MODIS value. The primary conclusion is that after assimilation of the MODIS data, the MATCH model remains closer to the mean MODIS values and the overall fit to the MODIS data is slightly better in MATCH where the linear slope is 0.66 for MATCH and 0.54 for MODIS.*

(The difference between MERRA-2 and MATCH AOD shown in Figure 4 is probably caused by aerosol models (we need to ask Fillmore about his thoughts on the reason for the difference). )

13. Figure 4 Line 261 why just 1 month (jan 2020) and not 15+years? It is interesting that AOD increase from clear-sky to all-sky in Match apparently only happens pollution regions over E. Asia US and Europe, which begs the question if assumed ambient rel. humidity aerosol water uptake in Match at higher rel.hum is too strong (also clouds with 100% hum. only occupy specific layers). And with clouds in the neighborhood there is not only the issue with aerosol size swelling but also the issue with wet removal. Thus, global models are highly diverse if AOD under all-sky are actually larger or smaller with respect to the clear-sky value. The large difference over biomass regions suggest that aerosol (fine-mode) absorption is much stronger in Match. Thus I would compare fine-mode SSA and water uptake between both models.

*We deal with only one month for this analysis as it is necessary to see the effect of clear sky on results. One cannot make a multiyear clear sky comparison. We attribute the increase in AOD from clear sky to all sky MATCH to the climatological aerosol used in the MATCH model. Clear sky results should generally correspond to time assimilated values matching more closely with the AOD from MODIS which should provide a more robust result.*

*We disagree with the reviewer's comment "The large difference over biomass regions suggest that aerosol (fine-mode) absorption is much stronger in Match". First, MATCH does not separate a fine mode from a coarse mode. Second, AOD is constrained by MODIS and Figure 4 shows AOD comparisons. SSA plays a minor role for the difference shown in Figure 4. Regarding MATCH having increased AOD over tropical forests relative to MERRA we state in the text that we suspect this is due to our lack of utilizing Quality Assurance values from the MODIS product. Whereas MERRA using the MODIS combined Deep Blue/Dark Target (where MODIS, when combining the two products, uses the QC parameters) MATCH assimilates both Deep Blue and Dark Target into its noon zonal band (as shown in Figure?) and we do not check the QC*

*parameters. We believe this allows for higher AOD's, possibly due to cloud masking errors in the individual products, that we are not filtering out, but are filtered out of MERRA. This will be corrected in Edition 5.*

14. Line 268 ... and aerosol absorption (and dust size)

*We do not do these other comparisons because of limitations described in the summary paragraph above.*

15. Figure 5 line 291, Bejing is an urban difficult site with a lot of near surface pollution during winter. Both models suggest larger AOD with clouds, whereby the increase is much stronger in Match than Merra. Otherwise the AERONET values are larger, most likely due to nitrate aerosol, which is missing in Match and possibly also Merra.

*We concur with the reviewer's comment.*

16. Line 304 the cases with less than 1% (in 1deg regions) are probably too few for useful statistics. (I would split all cases in ca equal counts with more and less clouds). I am bit surprised than with the 1% threshold relative many cases are found ... so do I misinterpret here? Also note that in the comparisons many AERONET sites are more local (urban) and not regionally representative.

*In fact the reviewers comment is pertinent. If we restrict AERONET observations to when an approximately 111Km2 grid box is completely free of clouds (as determined by CERES satellite cloud analyses) the sample is quite small, particularly over cloudy regions like the Amazon and Congo. But, by increasing the filter to 1% the numbers increase significantly giving more robust samples.*

*AERONET sites were chosen based on regional location and length of record. We did not consider rural vs. urban in our decision to include a site.*

17. Line 325 ... would this make you want to reduce the water uptake in Match?

*We do not know if the water uptake in MATCH is too large or simply MATCH produces too much aerosol. Our sense of the error as described in the discussion is that MATCH is too high compared with AERONET in summertime over Beijing.*

18. Table 2 Line 341 I wonder about the many matches for Australia, the small AOD values over Brazil with the biomass burning, the relatively large AOD values over central Africa (although the sites are spread. and the small values for N.Africa associated with dust. I strongly recommend to use AERONET as reference not just for AOD, but also for the split in AODf and AODc and also for AAOD data (at larger AERONET AOD cases AOD550 > 0.25). Given the large differences between the models in some regions (in Fig 4) the bias differences among the models are relatively small ... why?

*We have at our disposal no simple method to separate out fine vs. coarse mode aerosols from the output retained. Though we note that the distinction between fine and coarse is somewhat arbitrary as it is based solely on the observed AERONET size distribution. Thus a fine aerosol might be 0.1um at one site and 1.0um at another. (Figure 9 below, taken from Dubovik et al. 2000.)*

[Figure]

**Figure 9.** Volume size distributions of inhomogeneous aerosols (externally and internally mixed) retrieved using the model of scattering by homogeneous spheres.

*None the less we did attempt such a comparison simply using large and small dust particles as defined by MATCH and plot them here against fine and coarse mode observations from AERONET at two sites likely to be affected by dust, Ilorin, Nigeria and Izana, Tenerife, shown below.*

[Figure]

*AERONET Fine mode fraction of AOD compared to MATCH average of dust particles < 1um. At Ilorin there is some correlation and similarity in magnitude, but not at Izana.*

[Figure]

*AERONET Coarse mode fraction of AOD compared to MATCH average of dust particles >1um. At Ilorin there is some correlation and large disparity in magnitude while Izana, there is a good correlation and similar values.*

*The point of these plots is that the correlation between the Fine and Coarse mode AERONET observations with, in this case dust aerosols from MATCH, is somewhat random. True fine and coarse modes would require a detailed extraction of all the aerosol profile data, after water uptake by soluble aerosols had been taken into consideration. These values are not retained and would require significant code changes and re-running of the product.*

19. Line 440  Figure 10 (not 8

*Thank you for catching that mistake, it has been corrected.*

20. Line 469  also dust size is an issue (as larger dust for the same mid-vis AOD has lower SSA and also larger spectral near-IR AOD)

*This is true and as MATCH does not resolve size per se it is an issue for the MATCH/Langley Fu & Liou aerosol scattering properties as well.*

21. Line 511  I do not follow the argument of a likely lower dust AOD … as such a tendency seems to go the right direction … given the 0.07 bias in Table 2…??

*We state in this paragraph that the in order to better match observed TOA reflected SW up, the EBAF-surface product adjusts the AOD upward, on average, globally, by ~0.02. Further we state this seems reasonable for mid-latitude and Asian regions based on our comparisons with AERONET.  Lines 511 through 513 point out that this increase in AOD, however, is inconsistent with results shown in Table 2 for North Africa where we are already biased high relative to AERONET observations. We have changed the sentence to: "The positive bias found in the downward shortwave irradiance for the North Africa group (Fig 12c) is not consistent with the positive bias of aerosol optical depth shown in Table 2."*

22. Line 560  I wonder about the surface albedo treatment (at 14% solar broadband values are mentioned and now a significant 3% increase was mentioned). I hope the Kato model uses the spectral dependency for local soil conditions from MODIS. I would tinker not just with AOD but also with size and absorption before starting to change the surface albedo.

*The spectral albedo shape is derived from the MODIS surface albedo data product. However, we believe that the effect of surface albedo to downward shortwave irradiance under clear-sky conditions is small compared to the effect of AOD. The spectral shape of the albedo is not adjusted in the EBAF process, only the broadband albedo difference.*

Computing surface irradiance are with spectral albedo. Irradiance adjustment in the EBAF process uses only a broadband albedo difference.

23. Line 571  … exactly  … and NO it should be part of this study.

*While we appreciate the desire to see more results, we will have to leave this to a future study where we anticipate the variables needed and output them as the model is run.*

---

## Author Comment (AC2)

Dear Authors and Editor,

This is a well-written paper, presenting clearly an important subject. I enjoyed reading it and appreciated particularly the finding that the presence of clouds leads to increased observed AOD by clear-sky instruments. However, even though the manuscript is generally scientifically rigorous, towards the end (Section 4) it produces some unfounded and arbitrary results. I see that this has been pointed out also by another reviewer. In my view some parts should be either removed or preferably further explored. I think a major revision is necessary before proceeding with the publication process.

Moreover, very little is said on the MATCH assignment of aerosol types and its agreement with observations. I understand that you would like to address the aerosol types in another publication, but a brief presentation of dominant types by region would be useful here. Then the reader would be able to understand better the possible effects of the aerosol types on the fluxes.

*We would like to thank this reviewer for the close reading and interest in our paper. We appreciate the reviewer's comments, questions and suggestions and have incorporated several of the reviewers' comments along with our responses into the paper. However, we are not entirely sure which results the reviewer finds either unfounded or arbitrary. Within section 4 we discuss the discrepancy between clear sky calculations and observed surface irradiance. At line 523 we state clearly "…we cannot identify the cause of the discrepancy….", and subsequently propose possibilities. While again, perhaps not entirely satisfying from our or a reviewer's perspective it is encapsulates our current understanding of our results.*

*Regarding the second paragraph above we have expanded our description of the aerosol types in the MATCH model, their optical properties, and how those types and properties are transferred into the radiative transfer calculations. As to an analysis of aerosol type by region, our comparison of AOD to AERONET observations is broken down regionally which, generally identifies dominant aerosol types as stated in the paper. Both reviewers requested such an analysis and in our we show that given the information currently available to us, attempting to extract estimates of fine and coarse properties from MATCH is inconclusive at best. This is shown in the following discussion (which is repeated for both reviewers):*

*We have at our disposal no simple method to separate out fine vs. coarse mode aerosols from the output retained. Though we note that the distinction between fine and coarse is somewhat arbitrary as it is based solely on the observed AERONET size distribution. Thus, a fine aerosol might be 0.1um at one site and 1.0um at another. (Figure 9 below, taken from Dubovik et al. 2000.)*

[Figure]

**Figure 9.** Volume size distributions of inhomogeneous aerosols (externally and internally mixed) retrieved using the model of scattering by homogeneous spheres.

*None the less we did attempt such a comparison simply using large and small dust particles as defined by MATCH and plot them here against fine and coarse mode observations from AERONET at two sites likely to be affected by dust, Ilorin, Nigeria and Izana, Tenerife, shown below.*

[Figure]

*AERONET Fine mode fraction of AOD compared to MATCH average of dust particles < 1um. At Ilorin there is some correlation and similarity in magnitude, but not at Izana.*

[Figure]

*AERONET Coarse mode fraction of AOD compared to MATCH average of dust particles >1um. At Ilorin there is some correlation and large disparity in magnitude while Izana, there is a good correlation and similar values.*

*The point of these plots is that the correlation between the Fine and Coarse mode AERONET observations with, in this case dust aerosols from MATCH, is somewhat random. True fine and coarse modes would require a detailed extraction of all the aerosol profile data, after water uptake by soluble aerosols had been taken into consideration. These values are not retained and would require significant code changes and re-running of the product.*

Please see below for more specific comments (main and trivial intermixed and shown by "order of appearance")

1) Line 82: In this manuscript you say little on aerosol optical properties other than AOD. For example (*t*)here (*is*) no mention is made on the single scattering albedo (SSA) and asymmetry parameter g, or equivalently on aerosol types.

*For the aerosol types used in the running of the Langley Fu & Liou radiative transfer (RT)model we have added a plot (now Fig 1) of SSA and g at 550 nm. Hygroscopic aerosols are shown as a function of relative humidity. The relationship between the MATCH aerosol types and those used in the RT model is explained in line 150-160.*

2) Lines 106-107: "The MATCH model is described in Section 1. In Section 2, we explain the aerosol transport model briefly." MATCH is the aerosol transport model, and Section 1 is the introduction. Should the sentence "The MATCH model is described in Section 1" be removed?

*Thank you catching this mistake, the line has been removed.*

3) Line 124: The breakup of types is not the same as in Table 1. The division between soluble and insoluble is not explained, does it refer to organic particles? Is fine dust a different type than coarse dust? If yes, this opens possibilities for a later (here or in another publication) comparison of the two types to the AERONET fine and coarse dust.

*The reviewer is correct. That list of aerosol types represents those found in the Langley Fu & Liou radiative transfer model. The list has been changed to show those types and we have added discussion along with a table and plot to better define how aerosol properties are treated in the models. Table 1 shows the transition from MATCH to Langley Fu & Liou constituents and Figure 1 shows the distinction between insoluble aerosol (given as constants) and soluble (plotted as lines.)*

4) Lines 124-125: "Model physics... of soluble gases and aerosols." Two sentences are saying practically the same thing.

*We have changed the sentence to read "Model physics included in MATCH are parameterizations for convection and boundary layer processes that include…." to clarify that processes mentioned in the second half of the sentence detail the 'physics' mentioned in the first half of the sentence.*

5) Table 1 and lines 145-148: So carbonaceous and sulfate particles are climatological, while sea salt and dust are dynamically generated. This, together with the lines 126-127 saying that MATCH parameterizes "the scavenging of soluble species", confuses readers

*Table 1 states that the source of sulfate and carbon are climatological. Lines 126-17 and 145-148 state that those aerosol fields subsequently evolve through advection and other processes.*

6) Line 171: "we do not use a quality assurance confidence score". Later you claim that the use of QAC from MERRA in convective regions creates a bias (wrt MODIS) opposite in sign but similar to the bias from MATCH, supporting the view that QAC is not necessary. This argument is a little superficial. The cause of the different biases between MERRA2 and MATCH is not sufficiently attributed to the use of the QAC. Moreover, a better check for the usefulness of the QAC would be to compare MATCH results with and without low QAC values. I am not saying that the authors should do this, but merely ask for an explanation why the QAC scores are not used in MATCH.

*The failure to use the Quality Control flags was an oversight during the development of the code. It will be corrected in the Edition 5 data product release.*

7) Figure 1 and relevant text: If you want to show the effect of AOD assimilation, why show the differences between consecutive hours and not the differences between assimilation and non-assimilation runs? Are the latter not available?

*As the reviewer has surmised, the latter are not available.*

8) Figure 1 and relevant text: I guess that the highlighted 15 deg bands is where local noon occurs. Still, I am not sure why and how the assimilation is taking place there and then. Is it because local noon is between the 10:30 and 13:30 overpasses? Are the Terra and Aqua AOD averaged and assigned to local noon?

*We agree that the introductory sentence to the paragraph is not as descriptive of the process of assimilation as we had hoped. It has been replaced with the following:*

*"The assimilation process begins by combining the dark target and deep blue aerosol retrievals from MODIS (both Terra and Aqua when available) and creating global daily averages for the month. As MATCH progresses through time the data at local solar noon are assimilated by taking a 15° longitude width of retrieved AOD from the daily mean maps."*

9)  Lines 189-193: So if a volcano is captured by MODIS, does MATCH amplify AOD for all aerosol types? MERRA2 had the same problems for the Pinatubo eruption, with the aerosol speciation being wrong in the post-Pinatubo years.

*We only assimilate the MODIS AOD. The aerosol mass is then distributed spatially (and vertically) but the aerosol type is mapped (per Table 2) into the available species in the Langley Fu & Liou RT code. Subsequently, yes, the process developed here will amplify all aerosol types (and so speciation may be incorrect) defined by MATCH for a given grid box where the MODIS data is assimilated. We point out this fact in lines 201-205 (new version of paper.)*

10)  Line 199: MERRA 2 assimilates AOD also from AVHRR for your first two years of your comparison (1979-2002) and MISR over bright desert regions (2000-2014)

*Thank you. That information has been added. The sentence now reads: "MERRA2 also assimilates surface observed AOD by AEROENT and ship born AOD observations as well as AVHRR and MISR retrievals for the years 2000-2002 and 2000-2014 respectively."*

11)  Figure 2: Which MODIS product is MOD08Mdy08? I couldn't identify it anywhere. It has to be daily, L3, DT and DB combined, right?

*We use two MODIS products, the MOD08_D3 from Terra and MYDO8_D3 from Aqua. We have changed the plot to read simply "MERRA-2 – MODIS" and "MATCH – MODIS" and clarified the product definitions in the text.*

12) Lines 207-211. I would like some clarifications. MODIS AOD comes from the clear-sky pixels in its 1x1 cell. Each day with at least some clear-sky pixels in MODIS, the whole all-sky 1.9x1.9 MATCH and 1x1 MERRA-2 cells are used. So at each cell you are comparing the clear-sky MODIS data with the all-sky data from the whole cell, aren't you? You address the problem with the MATCH and MERRA2 daily average data vs. the MODIS only-overpass data, but if I understand correctly, the clear-sky MODIS vs. all-sky MATCH/MERRA2 problem would be present even for instantaneous values at the overpass times.

*When MODIS clear-sky AOD is available for a grid box, the assimilation will replace MATCH AOD regardless of the cloud fraction in the MATCH grid box at the time. For this analysis we create monthly average maps from daily mean MODIS AOD maps created for the assimilation process into the MATCH model. However, even with both Aqua and Terra data, daily mean MODIS maps are not complete due to clouds. (See image below showing a daily mean AOD map from Terra.) These clear sky MODIS daily averages are then used as a guide for when and where to sample the MERRA and MATCH models across a given month.*

[Figure]

*With 20 years of data, the maps are eventually filled completely though sampling does vary from grid box to grid box. Figure 2) shows the differences between the mean MODIS map and the MERRA2 and MATCH, sampled at the day/place where MODIS has values.*

13) Fig. 4 and relevant text: Again, I would like some clarifications. If clear-sky AOD is available, why not compare it against MODIS? Besides, how can you get clear-sky AOD from all-sky AOD by weighing with the clear fraction? Also, the mean MATCH clear-sky AOD in Fig 4bottom is 0.136, while the mean clear-sky AOD in Fig 3b is 0.160. Apparently, the term "clear-sky" means different things here, but I am not sure what is the difference exactly.

*The plots shown in bottom of Fig 4 (now Fig 5) are not true clear sky AOD. They represent an average, over 20 years, of monthly mean MATCH and MERRA2 AOD that used the fractional clear sky within a 1degree grid box as a weight for the averaging process. We have changed the text in lines 257-264 (new version of paper) and in the caption to Fig 5 and added the reference to Loeb et al. 2020 where this technique was explained.*

14) Fig. 4 and relevant text: In l. 244 you mention "March 2000 through February 2020", which also seems to agree with the color bar text. However, in the figure caption you mention "January 2020". Judging from the maps, the former seems to be correct, but please correct the inconsistency.

*Thank you for the close reading. These maps are 20-year climatologies as you note. 'January 2020' has been removed both within the picture caption and earlier in the text the term 'monthly mean' was replaced with the words 'climatological mean'.*

15) Lines 263-264: "MATCH ... larger... than MODIS". Even though this seems to be the case in Fig. 2b, Fig 3 gives the MATCH global average as 0.16 vs the MODIS 0.174. What am I missing? Area-weighted averaging?

*That comment was left over from a previous edit. The reviewer is correct that both are smaller than MODIS. We have corrected the sentence to read:*

"The above results indicate that both MATCH $AOD_{MODIS}^{clr}$ and MERRA2 $\overline{AOD}_{MODIS}^{clr}$ are generally smaller than MODIS $\overline{AOD}_{MODIS}^{clr}$ ."

16) Line 294: I guess the two channels are the 500 and 675 nm and interpolation is used to get the AOD at 550 nm.

*We use the angstrom exponent as given in AERONET data files to derive a value at 550nm, generally, from the 500 nm & 675 nm channels if available.*

17) Line 299: "45 sites". More than 45 sites are shown in Fig. 6 and Table 2. Also, Fig. 6 and Table 2 disagree in the numbers.

*Thank you for finding that error. The total number of AERONET sites used 55 as listed in Tables 2 and 3. The count has been corrected in the text and in Figure 6. (Revised manuscript Figure 7.)*

18) Line 316: I think it should read "MATCH clear-sky AOD for the Brazil group is biased high by 0.02"

*Thank you for finding that error. The text has been corrected to "0.02."*

19) Line 392: "fixed aerosol sources in time". This goes only for carbonaceous and sulfates, right?

*Correct. These types are listed as such in Table 1 as 'monthly climatological'.*

20) Line 440: I think it should read "scale nearly linearly with AOD between"

*Yes. We have added the words 'with AOD' to the sentence.*

21) Line 440: It should be Fig. 10 instead of 8

*Correct. Thanks for finding that error. Text has been changed accordingly.*

22) Line 468: Moreover, the polar irradiance is small, so it makes sense to see the smallest overestimations.

*That sentence has been removed.*

23) Line 469: Or overestimation of the asymmetry parameter? Or underestimation of ozone? Maybe a general bias of 1-2% for the Fu-Liu model? There is a known overestimation in the model-calculated surface SW fluxes, still under examination (e.g. https://doi.org/10.1007/s00382-018-4413-y). My point here is that the analysis is not deep enough to provide a definite answer and that these findings are not robust.

*The findings are robust in so much as the data available will allow. For that reason, we state in line 468: "This points to the possibility…". We do not make a definitive statement on the source of the error, only suggest a possible reason for the overestimation of DSW in the model. As the reviewer points out, there are other possibilities.*

24) Fig. 12: Instead of StdDev, it would be better to use RMSE, similarly to Fig. 13

*We agree with the reviewer in that it is good to be consistent so have replaced the figure with values of RMSE instead of standard deviation.*

25) Line 477: "increase the observed downward shortwave irradiances". Do you mean by reflection on the clouds?

*Yes, or any increase in the diffuse observation due to clouds not identified by the satellite. We have changed the text to be more specific, new version of the paper, lines 480-484.*

26) Line 493: Maybe include a brief description of the Kato et al. (2018) methodology for the tuning and possibly justification for these specific adjustments? Otherwise, these adjustments seem arbitrary. I understand that you are trying to make the atmosphere less transparent and more reflective and that the algorithm works by adjusting AOD, albedo and atmospheric water. However, the desired changes in both TOA and surface fluxes may be due to problems in aerosol misclassification, errors in the optical properties of each aerosol type, the neglect of aerosol vertical profiles, etc. If you cannot exclude these sources of error, I think it is premature to assign corrections to AOD, albedo, and water vapor to fit the CERES fluxes. It is useful as a sensitivity study, but its usefulness ends there.

*We understand the reviewer's concern. Water vapor affects TOA clear-sky shortwave flux. But it also affects TOA LW flux. Therefore, the adjustment of water vapor amount is constrained by longwave. Clear-sky shortwave adjustment is, therefore, dependent upon how we distribute the difference between surface albedo and aerosol properties. For ocean surfaces that occupy 70% of Earth, the uncertainty of surface albedo is small compared with the uncertainty in the aerosol properties, provided there is no sea ice and wind speed is known. TOA flux is sensitive to single scattering albedo and scaled optical thickness. However, the variability in the optical thickness is much larger than the variability in the single scattering albedo and asymmetry parameter. This justifies, to a first order, adjusting aerosol optical thickness over ocean. Note that we do not have vertical profile information because only passive sensors are used in CERES data production. To further clarify our intent, we have added the following to briefly describe the adjustment process:*
*"After known biases are taken out, the adjustment of temperature and specific humidity profiles, surface and aerosol properties are derived based on their pre-assigned uncertainty and the difference of computed and observed TOA shortwave and longwave irradiance using the Lagrange multiplier approach."*

27) Line 502: You probably mean Table 4

*The reviewer is correct it should be Table 4. (In the revised manuscript, now Table 5.)*

28) Line 510: You probably mean the top left plot of Fig. 12

*The reviewer is correct it should be Fig 12. (In the revised manuscript, now Fig 13.)*

29) Line 511: Here too, Fig. 12

*The reviewer is correct it should be Fig 12. (In the revised manuscript, now Fig 13.)*

30) Line 511: You write "decreasing AODs for the desert group by 0.02", but Table 4 shows that the AOD adjustment for the desert is +0.02

*This was poorly worded, and the paragraph has been updated. We state in this paragraph that in order to better match observed TOA reflected SW up, the EBAF-surface product adjusts the AOD upward, on average, globally, by ~0.02. Further we state this seems reasonable for mid-latitude and Asian regions based on our comparisons with AERONET (Table 2). Lines 511 through 513 point out that this increase in AOD, however, is inconsistent with results shown in Table 2 for North Africa where we are already biased high relative to AERONET observations. We have changed the sentence to:*
*"The positive bias found in the downward shortwave irradiance for the North Africa group (Fig 12c) is not consistent with the positive bias of aerosol optical depth shown in Table 2." (Lines 21-523, new version of paper.)*

31) Line 567: This is maybe subjective, but for clear-sky fluxes I would not call the role of aerosol optical properties minor. Definitely not for the AOD, but even for the SSA.

*We have removed the word 'minor'.*

---

## Referee Report (RR1)

Review of the revised paper:

**Evaluation of aerosol optical depths and clear-sky radiative fluxes of the CRERES Edition 4.1 SYN1deg data product**    *by D.Fillmore et al.*

**Comments to the author's responses**

All my comments were addressed – although those involving extra analyses were unfortunately denied - so if not in this paper … then hopefully in a future contribution. When the authors state as the goal "to evaluate aerosol optical depths for irradiance computations", then also the clear-sky LW fluxes need to addressed, especially near dust sources. Typical mineral dust aerosol radii near sources are 3-6um and even transported dust has effective radii of about 1.5um. Their given justification to ignore LW effects assumes too small (ca 0.5um) radii for mineral dust. In that context also the last sentence of the abstract ('… are unknown.') is very unsatisfying. For the dust inconsistency problem also possibly poor choices of AERONET references contribute, as (1) column properties (like AOD) of regions should not be compared to mountain site data (as Izana) and (2) months when other aerosol types dominate should be excluded (as for DJF in western Africa). And if AERONET is applied to reveal biases for AOD, then also the offered AOD split between fine-mode/coarse-mode AOD should be applied. I reject the notion that MATCH does not contain (approximately) this size information with its component processing, where dust and seasalt basically define the coarse mode AOD and (organic and black) carbon and sulfate define the fine-mode AOD. And for cases, where/when relevant AERONET sites are missing (as for northern Africa, central Africa or ocean regions), comparisons at coarser temporal (e.g. monthly, seasonal) resolutions (as offered by top down climatologies or other approaches such CERES, SRB 4, ICAP) are (as in my initial review) encouraged for insights on MATCH tendencies.

The strength of MATCH AOD data are high temporal resolution AOD maps (as needed in modeling) to include short-term regional anomalies, but that does not mean that global multi-annual (average) maps are correct. The most important section is chapter 4. So there should be more weight (and analysis) and less weight on the first 25 pages. In particular, the detailed comparisons to MERRA distract, as it is only a different model interpretation with quite different results and MERRA results are even left out in the comparisons at TOA fluxes (CERES) and surface fluxes (BRSN) of chapter 4.

On responses to my minor points of the initial review …
- The demonstrated smaller AERONET AOD at cloud-free conditions at the CART/Bondville sites may be meteorology (air-mass) related. These statistics will not necessarily apply to many other AERONET sites, for instance at dust dominated sites where higher AOD will likely be associated with less clouds.
- The assumed 1um mineral dust size (0.5um eff radius) is very small, even for transported dust. Thus, the associated reasoning for LW impacts are not important (to the surface irradiance) is not convincing at all. Also with larger (up to 10um) dust sizes the effective solar absorption potential (e.g. 1-SSA) quickly increases so that for the same AOD the solar surface irradiances will be lower (dust size in MATCH could be possibly included via a proportional link between dust AOD and dust effective size, as sizes near dust sources are usually larger).
- in Figure 1 there are not the promised maps for SSA and g ….?
- I agree with the authors and retract from my initial conclusion of a "likely stronger fine-mode absorption in MATCH compared to MERRA", because Figure 4 compares model simulated AOD maps. Still, the larger MATCH AOD values in regions, where fine-mode aerosol (SU, OC, BC) types dominate are concerning – possibly helped by the fact that MATCH assimilations only constrains the total AOD and not local component mixtures.
- I still suggest in AOD assessments to separate fine and coarse mode AOD contributions when

comparing to MERRA and AERONET. For AERONET inversions detailed (ambient) 22 bin size classes a separation at the 0.528um size-bin boundary is recommended and in modeling the separation by components with combining BC/OC/SU (fine) and SS/DU (coarse). AERONET also offers simple fine-mode fractions via the AOD spectral dependence (and the SDA method) from direct attenuation data. The fine-mode effective radius can certainly vary (as demonstrated in a figure by the authors) but there are always a minimum at aerosol radii of about 0.5um (unless there is a major volcanic eruption with effective sulfate aerosol radii near 0.5um).
- if you use high altitude mountain AERONET sites (e.g. Izana) then you get underestimates in comparisons to regional averages (bad choice!). Also West-African sites (e.g. Ilorin) are biomass dominated in NH winter and dust dominated in NH summer, which complicates a type association unless a seasonal separation is done. You also present fine-mode and coarse mode distributions but I suspect that fine-mode is not dust (as fine-mode dust is secondary to coarse mode dust).

**Comments to the responses**

36      these maybe convective regions … but a much more important element is that these are regions with fine-mode aerosol maxima (by wildfires and pollution).

45      make sure to pick regionally presentative AERONET dust site (exclude mountain sites)

47      if AOD is correct, but the dust size (and solar absorption) is underestimated, then a model yields too high solar irradiances at the surface

152      OPAC is parts in outdated (e.g. too much dust absorption, too little BC absorption) … but more importantly, how is OPAC applied? I assume (if so state that) that MODIS AOD are compared to simulated AOD and differences are expressed via component mass corrections by applying OPAC aerosol type based Mass Extinction Efficiencies – assuming locally/monthly fixed aerosol component mixtures.

158      Sinyuk (2003) dust RFI-imag values are much better than those in OPAC … also you may note that in his paper the coarse mode dust effective radius (in Table 1) is ca 1.5um (number mode radius of 0.63um and stddev of 1.72).

182      as from AOD differences aerosol component mass is adjusted … this of course requires mass extinction coefficients for aerosol type (which also include water uptake). Are these coefficients based on OPAC and if so have they been checked for realism?

205      errors in the captions: "thickens" and "left)" and "right)"

222      … and eastern EU / west Asia

230      I do not see that "Match-Modis differences are smaller than Merra-Modis differences". And there is no plausible explanation by MERRA is so much smaller than MODIS and by MATCH over many land regions is even larger than MODIS (based on Figure 2 differences I am surprised to see in Figure 3 that the global averages of MATCH is smaller than MODIS? Also in Figure 4 clear and all-sky MATCH-MODIS differences are positive, which is inconsistent with Figure 3 means).

265      I suggest to replace 'convective' with 'fine-mode AOD maxima'

283      Anomalous high AOD by MATCH at near overcast conditions (as shown in Figure 5) will not matter much for solar irradiances at the surface, when clouds reflect most of the solar energy… or? So for solar irradiance all-sky AODs (and 'averages') are less meaningful that clear-sky AODs.

340    Many aerosol types have if not monthly so at least seasonal maxima, so an evaluation to AERONET (in Table 2) on a seasonal basis would be much more insightful … even better with a separation into fine-mode and coarse-mode AOD.

387    … so what is the conclusion? Spell it out. MODIS AOD are likely overestimated without QAC so that MATCH is relatively high compared to MERRA. But this does not answer the question why MATCH is significant larger than MODIS over many continental regions as shown in Figure 2.

393    If MATCH (via MODIS) misses large aerosol events, why are AOD values over continents (incl. wildfire regions) so large?

422    the AOD data presented in Figure 9 are maxima (if I read the captions correctly). Why are not median and interquartile-range values are presented instead? And is there a direct sample link between AOD and PV of just max statistics?

426    explain 100% post and neg error in the captions. If the guess is 0.2 … then at 100% the lower bound is 0.1 and the upper bound 0.4, which would give an asymmetry not shown in Figure 10. Also it would be nice to show results for a more absorbing (dust or wildfire) aerosol, because the error max at sza near 0.5 will vanish and will more simply decrease with insolation. Still I do not see the value of this figure as not even an AOD value is given in the captions (as usually AOD uncertainty wants to be translated into W/m2 irradiance error). Why do I have to read the text to get the info and with those definitions even negative AOD are possible..? Figures and captions should be self-explanatory! But do we need section 3.1 other than stating AOD error as function of aerosol type and SZA… all of which could be placed in a table.

486    CERES detects SW (and LW) upward radiation (and not solar 'irradiances' as at the surface).

493    Aside from AOD and precipitable water there are other knobs for tuning such as (aerosol absorption, aerosol (dust) size and aerosol elevation, surface albedo). Just tuning with global averages may provide globally the correct result for the wrong regional/seasonal regions. Thus a regional evaluation is strongly encouraged. Later surface albedo changes are included … so are these not elements in the global adjustments, as they seem to be higher everywhere in Table 4

500    to increase the shortwave reflection in terms of aerosol properties also less aerosol absorption and a smaller aerosol type (relative more fine-mode AOD) helps

508    MATCH AOD is already high at mid-latitudes … even to MODIS, so an additional 0.02 AOD seems to go the wrong way.

510    there is no 'top right' in Figure 10

519    MODIS AOD over oceans are already likely too high, so that further increase can not be justified.

534    I miss in Table 4 the clear-sky upward MATCH based SW (and LW) uncorrected upward fluxes (to compare to CERES based TOA obs) and the clear-sky downward MATCH based SW (and LW) uncorrected downward fluxes (to compare to BSRN/buoy data).

---

## Referee Report (RR2)

Review of re-revised paper:

**Evaluation of aerosol optical depths and clear-sky radiative fluxes of the CERES Edition 4.1 SYN1deg data product _by D. Fillmore et al._**

**Highlights**
- needed paper as MATCH aerosol data are an important element CERES products
- now different larger dust sizes are included (good)    … but not correct SSAs (bad)

**Concerns**
- solar flux validation for aerosol cannot be drawn from AOD data alone – aerosol absorption and aerosol size are also important.
- apparent effect of decreasing SSA at increasing coarse dust sizes is ignored → wrong absorption is assumed

**General comments**

I have now reviewed the paper several times. This time there were no responses to my latest comments … so I just re-read its content. I am delighted that now larger dust size are considered (great), but unfortunately the associated lower SSA (with larger sizes – for the same RFimag) are ignored. This becomes quite apparent from the supplied simulations in Table 6. I have provided similar simulations, with the associated stronger AAOD (from lower SSA values) at larger sizes and I could demonstrate that for the particular set-up solar losses to the downward surface fluxes are twice as large with a dust size increase by a factor of 4. Thus, with a stronger mineral dust absorption, it is also likely that the' unresolved' issue for dust regions will disappear.

Thus, please redo your dust simulations and re-evaluate the dust area flux comparisons before publication.

**Specific comments**

37/38    Clarify this sentence: both Merra and Match are assimilations and the MODIS AOD input is the same. If not explain. Or has this to do that Merra allows for washout processes?

39-42    I assume that refers to Match AOD. So, the surface-flux comparison indicates solar attenuation is too small (-> AOD underestimates and/or absorption underestimates

42-45    So the CERES clear-sky reflection needs more reflection of the surface site (-> AOD overestimates and/or absorption underestimates.

… so if we combine the two findings… then the only conclusion is that MATCH aerosol absorption is underestimated (which for dust is likely an underestimation in size)

46/47    leaving issues unexplained (even in the abstract) is discouraging

167    now several dust sizes are allowed (great !!)

174    the SSA (and ASY) varies with large dust size (and even spectrally), update Figure 1 and simulations! For mineral dust re of 1.5, 2.5, 4.0, 6.5 and 10um the mid-visible SSA values are 0.962, 0.931, 0.918, .882 and .840 for the same imaginary part (here 0.0011). In addition, since the dust solar spectral Rfimag are larger towards the UV the SSA value at shorter wavelengths

are even lower (more absorbing)

223    just curious… are the land-sea contrast offsets of northern Africa a MODIS or a modeling problem? (I do not see very strong offsets in MODIS statistics.)

232    Merrra apparently includes other AOD data from other sources (MISR, AERONET), but MODIS data should dominate (in volume) so I would not expect so significant differences as displayed in Figure 3. How can you exclude that model-specific aerosol processing in the base-line model (without assimilations) is not the issue. This could be easily verified … or?

259    the least square fits and rms values (in figure 4) are dominated by the largest AOD, but not by the most frequent AOD, thus possibly also show the scatter plots in log/log scale

459    yes! … and remember a larger coarse dust size can also increase the aerosol absorption

489    this table is interesting and quite revealing. To demonstrate, I did similar simulations, using a solar zenith angle of 0.95 (not 1, oh well but close), a desert surface albedo, a mid-lat summer atmosphere, and dust with (only) a mid-visible optical depth of 0.2 for mineral dust located between 1 and 3 km altitude. Here different dust sizes with their associated SSA are applied (all dust size-distributions assume the same spectral refractive indices with RFimag at 550nm = 0.0011). The downward shortwave and longwave fluxes of these simulations are

|            |             | shortwave | longwave |
|------------|-------------|-----------|----------|
| No dust    |             | 998       | 346      |
| Dust (0.2) | reff= 1.5um | 985       | 354      |
| Dust (0.2) | reff= 2.5um | 983       | 357      |
| Dust (0.2) | reff= 4.0um | 980       | 359      |
| Dust (0.2) | reff= 6.5um | 974       | 359      |
| Dust (0.2) | reff= 10.um | 968       | 359      |

These calculations consider the lower SSA at larger size (as they should) and the solar losses at the surface flux losses for factor 4 dust size increase with these calculations are:
-15 W/m2  (between reff 10 and reff 2.5)  while the author's simulations - even with larger u0 (=1)
- 8 W/m2  (between reff 8 and reff 2).  So please redo your calculations with the correct (lower) ssa values at larger sizes … and you will have an explanation to your dust bias.

And the LW dust impact on downward fluxes depends strongly on the assumed dust vertical distribution as much as on size. Thus, all size (… from AeroNet inversions?), correct RFimag (especially in the stronger absorbing 8-10um region … from I.Sokolik?) and altitude (…from Calipso?) have to all accurate for useful clear-sky dn LW flux comparisons at the surface.

So when the correct dust sizes/SSAs are applied with the result of a stronger aerosol absorption possibly (with the right dust sizes, and right dust altitudes) correction to AOD and water vapor may not be necessary to bring SW and LW fluxes into agreement.

577  more clouds → more AOD? … not for dust, when clouds remove dust

593  redo your large dust SSA calculations … and the solar surface vs toa flux difference problem for mineral dust should be gone.

Below I attach a summary of the top-down (optics → component) approach of the MAC climatology, which lists in Table A1 different dust types, in Figure A3 (column3) seasonal averages of dust size (based on AAODc and AODc-DU), assumed size-distributions and spectral refractive indices in Figure A6 and resulting single scattering properties in Figure A7 (where 'DU+' is for larger dust with reff =4um and 'DU' is for background dust with reff=1.5um).

The Max-Planck Aerosol Climatology (MAC) offers merged monthly global maps for aerosol optical properties. In the merging process, multi-annual observational statistics of photometry from the ground is forced on spatial context supplied aerosol component 'bottom-up' modeling. The merged aerosol optical properties focus on aerosol column amount and aerosol column absorption, separately for smaller 'fine-mode' aerosol and larger 'coarse-mode' aerosols. In Figure A1, multi-annual averages from photometry (AERONET/MAN with absorption data only over continents) are compared to multi-(AeroCom phase 3) model interquartile averages (AC3-iqa). In addition, in that Figure A1 the resulting MAC version 3 maps are presented along with applied regional % changes to AC3-iqa.

[Figure]

**Figure A1.** *Global annual average distributions for AODf, AAODf, AODc and AAODc of (1) AERONET/MAN photometry (upper left), (2) the AeroCom phase III multi-model interquartile average background (AC3-iqa, upper right) and (3) resulting MAC climatology fields (lower right) - after AERONET/MAN adjustments in % to AC3-iqa (lower left). Numbers at the lower left of each image indicate global averages. Also note that absolute aerosol absorption data (AAOD) are multiplied by 10 to match the common scale.*

Major adjustments to the 'bottom-up' modeling background are strong increases to the 'fine-mode' absorption (AAODf) and decreases to the 'fine-mode' aerosol amount (AODf) at mid-latitudes. Also SH continental dust (AODc) is stronger. Absolute MAC version 3 changes with respect to both

background (AC3-iqa) and to the older version2 of the MAC climatology (*Kinne, 2019a*) are presented in Figures A2 and A3.

[Figure]

**Figure A2.** *Absolute annual difference global distributions for AODf, AAODf, AODc and AAODc between the MAC (v3) climatology and the AC3-iqa background (left block) or the older version (v2) of MAC (right block). Average differences are summarized by a value and for the common scale absorption data differences were multiplied by 10.*

Compared to the previous MAC (v2) version (*Kinne 2019a*), the new MAC (v3) climatology version uses and more recent (better emission data applying and component mixture permitting) modeling AC3-iqa background (instead of the AC1-median) and considers 238 (instead of 21) sub-regions for regional adjustments. This resulted in different fine-mode distributions over E. Asia on strongly increased fine-mode contributions over central Africa.

In a 'top-down' approach aerosol column and absorption properties are associated with a mixture of pre-defined aerosol components, that differ in size and mid-visible absorption. All MACv3 types are listed in Table 1.

**Table A1.** *size and absorption potential of pre-defined aerosol types in MACv3. Listed are (1) effective radii (Re) of log-normal distributions with associated mode radii (Rm) and standard deviations (sd), (2) mid-visible refractive indices (and from size and RF,imag resulting SSA values) and (3) column numbers (N) based on component global mid-visible AOD averages. Five different sizes are allowed for sulfate and dust, each. For comparison, values of a cumulus water cloud (water) and a cirrus cloud (ice) are provided.*

| aerosol type | label | Re | Rm | $s_d$ | $RF_R$ | $RF_I$ | SSA | < OD > | N |
|---|---|---|---|---|---|---|---|---|---|
| | | [um] | [um] | | \multicolumn{4}{c}{at 550nm wavelength} | | | | [#/m2] |
| **soot** (not used) | BC | .06 | .03 | 1.7 | 1.70 | .7000 | .155 | 0.005 | 4.0 e+11 |
| **soot + o.shell** | **BO** | **.12** | **.08** | **1.5** | | | **.615** | **0.015** | **4.0 e+11** |
| **organic** | **OC** | **.18** | **.12** | **1.5** | **1.53** | **.0050** | **.975** | **0.022** | **1.8 e+11** |
| sulfate | SU | .06 | .03 | 1.7 | 1.43 | .0000 | .999 | 0.023 | 4.4 e+13 |
| sulfate | SU | .10 | .05 | 1.7 | 1.43 | .0000 | .999 | 0.023 | 4.1 e+12 |
| **sulfate** | **SU** | **.16** | **.08** | **1.7** | **1.43** | **.0000** | **.999** | **0.023** | **6.0 e+11** |
| sulfate | SU | .26 | .13 | 1.7 | 1.43 | .0000 | .999 | 0.023 | 1.2 e+11 |
| sulfate | SU | .40 | .20 | 1.7 | 1.43 | .0000 | .999 | 0.023 | 3.8 e+10 |

| | | | | | | | | | |
|---|---|---|---|---|---|---|---|---|---|
| **seasalt** | **SS** | **2.5** | **.75** | **2.0** | **1.50** | **.0000** | **.999** | **0.035** | **3.3 e+09** |
| **dust** | **DU** | **1.5** | **0.93** | **1.55** | **1.53** | **.0011** | **.962** | **0.025** | **2.7 e+09** |
| dust | DU | 2.5 | 1.34 | 1.7 | 1.53 | .0011 | .931 | 0.025 | 1.3 e+09 |
| dust | DU | 4.0 | 1.55 | 1.85 | 1.53 | .0011 | .918 | 0.025 | 7.0 e+08 |
| dust | DU | 6.5 | 1.98 | 2.00 | 1.53 | .0011 | .882 | 0.025 | 3.6 e+08 |
| dust | DU | 10 | 2.30 | 2.15 | 1.53 | .0011 | .840 | 0.025 | 2.0 e+08 |
| cloud **water** | water | 10 | 6.7 | 1.5 | 1.33 | .0000 | .999 | 10.0 | 2.5 e+10 |
| cloud **ice** | Ice | 40 | 20 | 1.7 | 1.31 | .0000 | .999 | 0.5 | 1.1 e+08 |

For smaller fine-mode sizes (1) a strongly absorbing BO (a soot coated by organic material) type, (2) a weakly absorbing organic matter (OC) type and (3) a non-absorbing sulfate (SU) type are considered. Thus, the SU type includes other non-absorbing fine-mode contributions, as from nitrate or seasalt. In order, to separate absorption contributions between BO and OM, the BO/(BO+OM) ratios for AOD from AeroCom 'bottom-up' modeling are applied (i.e. BO/(BO+OM) AOD ratios are higher over urban pollution than over wildfire regions). While BO (Re=0.12um, with Re=0.06um BC cores) and OM (Re=0.18um) sizes are fixed, the SU size (0.06 < Re < 0.40um) remains variable to match fine-mode effective radii of the MAC climatology.

For larger coarse-mode sizes (1) a non-absorbing seasalt (SS) type and (2) an absorbing mineral dust (DU) type, are assumed. While the SS (Re=2.5um) size is fixed, the DU size (1.5 < Re <10um) is assumed to increase with coarse-mode DU-AOD. Note that even with a constant mid-visible imaginary part for dust – here assumed at 0.0011 (*Di Biagio, 2019*) – the mid-visible absorption potential $(1-SSA_{DU})$ increases with dust size. With an initial guess for the SS-AOD to extract the AODc associated with mineral dust, the following relationship is applied:

$$1-SSA_{DU} = 1-SSA_{DU,min} + 0.05* \text{ DU-AOD}$$

$1-SSA_{DU,min}$      = 0.037   (for the smallest assumed dust aerosol radius of 1.5um)
DU-AOD      = (AODc – SS-AOD,guess)
SS-AOD,guess      = .003*windspeed$_{SUR}$ [m/s] *(2-cos (2* (lat[deg]-sun[deg])/2)
     *(ocean_fraction)

As the coarse-mode absorption now defines both the DU-AOD and the size for dust, and the remaining coarse-mode AOD is assigned to seasalt to replace the initial SS-AOD guess.

Figure A3 presents seasonal averages for 'top-down' size-choices for sulfate and dust components, for the fine-mode Re of MAC and for the applied BO/(BO+OM) ratios from modeling.

[Figure]

**Figure A3.** *MACv3 seasonal maps for fine-mode Re (col1) and as part of the 'top-down' approach sizes (Re) for non-absorbing fine-mode (col2) and mineral dust (col3 – times 0.1, to fit common scale). Also presented are applied BO/(BO+OM) ratios from modeling (col4) to separate BO and OM. Numbers next to plots show seasonal averages.*

The resulting aerosol component AOD maps attributed in the 'top-down' approach are presented Figure A4: seasalt (SS) and dust (DU) from AODc and sulfate (SU), organic carbon (OC) and black carbon from AODf.

[Figure]

**Figure A4.** *MACv3 annual maps of 'top-down' derived AOD component distributions. The coarse mode AOD is split between seasalt (SS) and mineral dust (DU) contributions (left column). The fine-mode AOD is separated into non-absorbing sulfate (SU), strongly absorbing BC (multiplied by 10) and weakly absorbing OC. In a different split for total carbon (CA =OC+BC) contributions without soot (OM) and contributions containing a soot core (BO) are separated. Values next to the maps indicate global averages.*

The annual maps of Figure A4 are a subset of maps shown in Figure 2. To illustrate MAC updates with this version 3 annual difference maps to the previous version 2 (Kinne, 2019a) maps are presented in Figure A5. Global coarse-mode AOD contributions remained stable but DU-AOD is smaller (as well as maximum DU sizes) and SS-AOD is larger. Reduced global fine-mode AOD contributions are caused by significant SU-AOD reductions despite increases to OC-AOD and BC-AOD. Major regional component AOD differences between v 3 and v 2 of MAC are:

| | | |
|---|---|---|
| **DU** | - 0.005 | more DU over Arab waters, W. Africa, Patagonia, less DU over N. S. Am. and N. Africa |
| **SS** | +0.007 | more SS over mid-latitude oceans |
| **SU** | - 0.015 | more SU over N. India and central Africa, less SU over E. Asia, E. Europe, E. US, tropics |
| **OC** | +0.010 | more OC over central Africa, N. India, SE. Asia, S. America |
| **BC** | +0.001 | more BC over central Africa, N. India, northern hemisphere |

As the fine-mode AOD is smaller, also the anthropogenic (fine-mode) AOD is smaller and less absorbing with the sharp reduction to (scattering) SU-AOD. Also, the anthropogenic (coarse-mode) dust AOD is smaller.

[Figure]

**Figure A5.** *Absolute annual differences for 'top-down' component AOD data between the current MAC version3 (see Figure 2) and the previous version 2. Values next to the plots indicate global average differences.*

The big advantage of separating AODc and AODf in aerosol components which are completely defined by size (-distribution), composition (with its known refractive indices over the entire spectral solar and infrared spectral region) and shape (here spheres are assumed) is, that all three spectrally varying single scattering properties (as input for broadband radiative transfer simulations) are quickly calculated. This is done via (MIE-) scattering simulations for (1) extinction (EXT, attenuation per distance), (2) single scattering albedo (SSA, the scattering potential) and (3) asymmetry-factor (ASY, approximating the scattering distribution). These single scattering simulations can be done for every desired spectral (radiative transfer) model resolution, as long as the component refractive indices at that resolution are provided. The presented spectral (8 solar and 12 infrared) choices in Figures A6 and A7 below refer to the spectral resolution of the a subsequently used radiative transfer model. The addressed aerosol components in Figure A6 and A7, which were already introduced in Table 1, are soot (BC, Re=0.06um), an organic mixture with a soot core (BO, Re=0.12um), organic material (OC, Re=0.18um), sulfate (SU, Re=0.16um), seasalt (SS, Re=2.5um) and mineral dust (DU, Re=1.5). In addition, properties of a larger mineral dust size (DU+, Re=6.5um) and for a general comparisons also properties for a water cloud (water, Re=10um) and for an ice-cloud (ice, Re=40um) are included. In Figure A6, size-distributions and component refractive indices

are compared. Note, that for the component shell/core mixture of the BO component no combined refractive index is offered (internally calculated) and that for mineral dust and sulfate, independent of a selected aerosol size, the same refractive indices apply.

[Figure]

*Figure A6. Aerosol size distributions (left images) for pre-defined aerosol components (to match in presented concentrations the global average MACv3 AOD) and real and imaginary parts of the refractive indices (right images). Refractive indices are compared at central wavelengths of 8 solar and 12 infrared spectral bands. For comparisons, the size distribution and refractive indices for a cumulus cloud and for a cirrostratus cloud are shown.*

Mie simulation (assuming spherical aerosol shapes) then yield the single scattering properties (EXT, SSA, ASY), which are presented for the components of Figure A6 in Figure A7.

[Figure]

***Figure A7.*** *Calculated spectrally varying component single scattering properties for extinction (left - via the ratios to the component extinction at 550nm), for single scattering albedo (center) and for the asymmetry factor (right).*

According to AOD maps of Figure A4, component single scattering properties are combined (AOD is additive, SSA is weighted by AOD and ASY is weighted by AOD*SSA) to yield global maps. Resulting single scattering properties at four selected wavelengths for fine-mode and coarse mode aerosol are presented in Figure A8.

[Figure]

***Figure A8.*** *Component combined annual average single scattering properties maps (AOD – left col., SSA – center col., ASY – right col.) at four selected wavelengths: at .45, .55, 1.0 and 1.6um for the fine-mode AOD (left) and at .45, .55, 1um and 10um for the coarse-mode AOD (right). Numbers at the lower left indicate global annual averages.*

Note that for the fine-mode, the aerosol sizes are too small to yield significant radiative infrared effects (at wavelengths >4um), so that data are presented at another near-IR wavelength (1.6um - rather than 10um). Similarly to Figure A8, the resulting single scattering properties at four selected wavelengths for (fine-mode) anthropogenic aerosol and for total (fine-mode and coarse-mode combined) aerosol are presented in Figure A9.

[Figure]

***Figure A9.*** *Component combined annual average single scattering properties maps (AOD – left col., SSA – center col., ASY – right col.) at four selected wavelengths: at .45, .55, 1.0 and 1.6um for anthropogenic*

*(fine-mode) AOD (left) and at .45, .55, 1um and 10um for the total AOD (right). Lower left numbers indicate global annual averages.*

Now all aerosol needed optical properties are defined, so that radiative transfer simulation can be performed to determine the aerosol radiative effects.

---

## Editor Decision (ED1)

**Response to latest comments by the authors of **Evaluation of aerosol optical depths and clear-sky radiative fluxes of the CERES Edition 4.1 SYN1deg data product**

**General comments**

My comments are based on the recent responses by the authors, I received by e-mail. The authors simulated in sensitivity studies the dust treatment (for direct comparisons to my offered simulations), to demonstrate consistency and importance, which is appreciated. Still, it is apparent that in the described MATCH model, dust size underestimates (despite dust RFimag overestimates) may cause likely biases and at least offer a possible answer to the 'unresolved' bias issues. Aside from the dust-issue though, there are still larger differences between MATCH and MERRA which at least should receive some attention – as both models assimilated MODIS AOD data, which would imply a better agreement than demonstrated. So make sure that critical deficiencies and differences are well explained.
As the MATCH data output is used in the CERES (surface) flux products, it is important that context on assumptions - also via this paper - to the applied AOD values are revealed, independent of their accuracy. Thus, I do want to not to hold up a publication any further.

**Specific responses**

*Based on your comments below regarding the dependence of irradiance on particle size, specifically Table 6 in the paper, we have found that one SW number was incorrect. Once the correct number was in place, we find that we do not need to re-evaluate our calculations as they agree with the values provided in your review. While, at the same time we also agree that the SSA in the OPAC tables being used for dust particles likely underestimate SW absorption.*

Good to hear that the corrected mid-visible SSA for dust are now more consistent with my simulations (although more to the fact that underestimates in size is compensated by overestimates in dust imaginary part) – see below. As severe underestimates in dust size remain, I would worry about mismatches for longwave radiation (no size → no LW extinction)

*'The difference is largely due to the differing methods of assimilating the MODIS AOD data product and the use of quality flags in our assimilation.'*

*We believe it (the unusual land-sea contrast) is the MATCH model (and so 'base-line model assimilation') that is one of the primary sources of the differences. We state as much in lines 234-235.*

Are there arguments why MATCH or why MERRA assimilations should be trusted more … as the resulting AOD maps are so different? Does it simply mean that the influence of the baseline model (e.g. CAMS vs GOCART) is much stronger than the impact of the AOD data assimilations? That would be an important 'help/issue' to any manuscript reader.

It is a bit puzzling to me that you say that the 'undesired' land-sea contrast is likely associated with the MATCH model – as models are not expected to show these inconsistencies. It would be nice to get some more insights as to the reasons .. why?.

*MATCH optical thicknesses over desert sites for clear- and all-sky conditions are larger (Tables 3 and 4). Computed downward shortwave from North Africa groups is larger than observed downward shortwave irradiance at the surface. As the reviewer suggested, it appears that we underestimate shortwave absorption, if particle size is larger. Here are the single scattering albedos as a function of dust particle size with OPAC refractive index. These single scattering albedos are low compared with those provided by the reviewer.*
*Single scattering albedo of dust particles with OPAC refractive index*
*Particle size 1.9 (2.15) 0.78 (2.0) 0.39 (2.0)*

Is this particle size and diameter? Then this refers to reff of 0.95, 0.39 and 0.19um? This is way too small for dust.

*546 nm 6.67931E-01 8.00413E-01 8.78470E-01*
*642 nm 7.15135E-01 8.46123E-01 9.11902E-01*
*842 nm 7.63865E-01 8.85041E-01 9.37848E-01*
*1226 nm 7.84489E-01 9.02001E-01 9.47219E-01*

Given the listed spectral SSA at these small reff sizes in MATCH I conclude from comparisons to my data that the assumed (and likely outdated) imaginary part in MATCH is much larger than my mid-visible value of 0.0011. So there is a compensating effect for dust SSA but not for dust size. Speaking of size, I found in an older paper-version (sorry I did not have access to the current version of the paper) this table, where the use of larger dust sizes was reported (see below). This needs some clarification.

Table 2. Mapping of MATCH aerosol types into Radiative Transfer code.

| MATCH Constituent | Langley Fu & Liou Constituent | Langley Fu & Liou Spectral Properties |
|---|---|---|
| Sea Salt | Maritime | d'Almeida 1991 |
| Hydrophobic Organic Carbon | Insoluble | OPAC |
| Hydrophilic Black Carbon
Hydrophobic Black Carbon | Soot | OPAC |
| Hydrophilic Organic Carbon
Tropospheric Sulfate | Water Soluble (WASO) | OPAC |
| Volcanic
Stratospheric Sulfate | Suspended Organic (SUSO) | OPAC |
| Dust < 1.0μm | "Small" Dust | Sinyuk et al. (2003) |
| Dust 1.0 -2.5μm
Dust 2.5-5.0 μm
Dust 5.0-10.0 μm | "Large" Dust | Sinyuk et al. (2003) |

*We believe that the dust size the reviewer mentioned in his comment on line 174 is too large. But we get similar single scattering albedos with a smaller size using OPAC refractive indices.*

*Figure 1 shows the values currently in the radiative transfer model and so represent the calculations in the SYN1deg data product. Changing the values is not currently an option for the operational SYN1deg product.*

The given dust sizes in my 174 comment were only given to demonstrate the size-effect on SSA for mineral dust. Typical dust sizes have a reff near 1.5um (so in that context the authors are correct with their 'too large' statement). However in regions around dust sources, mineral dust aerosol sizes will be larger (at least reff=3 and up to reff=6). I am still worried about the smaller size options in MATCH. If and where (the authors should know) these are applied, dust absorption is likely underestimated. Maybe this can help in their bias assessments.

*We have changed the sentence to suggest the error may be due to the choice of dust particle size and distributions.*

Thanks, such a statement is much more satisfying.

*The log density plot does show the vast majority of AODs are less than ~0.6. And though the fit line is 'pulled down' somewhat by the larger AODs we feel a log/log plot on top of the log density would not significantly change the results presented. It might bring the fit line a little closer to the 1-1 line, but the point that both MERRA2 and MATCH underestimate AOD relative to MODIS will not change. It is also re-iterated in the statistics below the plots.*

The biases vary with AOD ranges (usually satellite remote sensing suggest larger (and likely overestimates) AOD at low AOD values, while AOD maxima are often missed) so possibly a separate linear plot for just the 0-0.3 AOD range could be an useful extra.

*First. we'd like to point out that there was in fact an error in Table 6. The value for Downward SW irradiance for particle size 2.0µm should have been 1038Wm-2, not 1028Wm-2. This has been corrected in the paper, line 489. This then brings our table values into line with the values presented above by the reviewer. Specifically, he records, for a fourfold increase in particle size, a decrease in DSI from 983Wm-2 to 968Wm-2 (~16Wm-2 or -1.5%). With the correct value in Table 6, a fourfold increase in particle size (2.0µm to 8µm) the DSI in our calculations decreases from 1038Wm-2 to 1020Wm-2 (~18Wm-2 or 1.7%). If we change our cos(SZA) to 0.95 (as the reviewer used) the DSI values at 2.0µm and 8µm are 977Wm-2 and 960Wm-2. So again, a similar decrease of ~17Wm-2 but the same percentage of 1.7%*

Thanks for checking/confirming

*The longwave numbers are correct in Table 6. To comment on the reviewer's statement that 'LW dust impact on downward fluxes depends strongly on the assumed dust vertical distribution as much as on size', in our offline radiative transfer model we increased the scale height of 1.5km to 5.0km for the same inputs as shown in Table 6. This resulted in DLI values at 0.5µm, 2.0µm and 8.0µm particle sizes of 351Wm-2, 357Wm-2 and 359Wm-2 respectively. Thus, increasing the scale height of the dust particles did not change the DLI by more than 1%, less than the changes due to particle size. This of course kept the particle size the same throughout the column which does not account for the fact that smaller dust particles are more likely to be lofted higher in the atmosphere.*

Thanks for the detailed answer, also working with 'my' relatively large sizes for dust. Unfortunately, these larger dust sizes are not considered in MATCH (correct me if I am wrong) so my concern was mainly that with max allowed dust diameters of 1.9um there would be no dust associated longwave effects as with larger dust sizes of the sensitivity simulations – mainly to TOA fluxes (where altitude placement matters) but also to longwave surface fluxes (where altitude placement is less of an issues – but opposite to TOA as: the lower → the stronger).

*we imply that more clouds indicate more AOD in the MODIS AOD retrieval process as*
*reported in the Varnai et al, 2017 reference.*

You are right, that in case of more (near-by) clouds MODIS retrievals tends to overestimate AOD but the reasons are less apparent. Most overestimates (over cloudy mid-lat. oceans) though are linked to fine-mode AOD so things are more complicated than simple cloud contamination. In case of modeling, things are less clear, as removal processes via clouds may dominate aerosol swelling effects.

*At this point in the processing of the MATCH model and its subsequent use in the SYN1deg radiative*
*transfer calculations, we cannot 'redo our large dust SSA calculations'. What we have done, and at this*
*reviewer's suggestion is try to show, in the paper, potential error due to the constraints we currently have*
*on our dust models in the radiative transfer code (Table 6) and the MATCH model's ability to define large*
*and small dust, Figures 15, 16 and related discussion.*

I understand that for the current MATCH version the SSA cannot be redone. So if potential MATCH issues it stated that should suffice for now.

---

## Author Response (AR2)

*Once again, we wish to thank these two reviewers for their interest and close reading of our paper. Based on suggestions by Reviewer 1 we have reorganized Section 4 of the paper, dividing it into three parts. Section 4.1 now addresses the impact of AOD on surface SW calculations. We added Section 4.2 which deals with LW surface calculations and the dependence of the LW and SW surface irradiance on dust particle size. (Section 4.2 includes a new table and three new images. To reduce the paper's length, we replaced Figure 14 with a much smaller table as suggested by Reviewer-1.) This new section allowed a segue into a brief discussion of LW aerosol effect and a comparison between observed and MATCH AOD as a function of fine/coarse particle sizes in terms of canonical monthly means. Regarding temporal variability and MATCH's ability to retrieve such, we focused on three desert sites where dust is found seasonally.  All substantial changes within the manuscript are highlighted in red. We believe we have answered the reviewer's questions and concerns, and this has resulted in a greatly improved work. Included in this file are our responses (in red) to both reviewer's queries, reviewer 1 first and reviewer 2 second.*
* * *
*Reviewer – 1*

Review of the revised paper:

**Evaluation of aerosol optical depths and clear-sky radiative fluxes of the CRERES Edition 4.1 SYN1deg data product** *by D.Fillmore et al.*

**Comments to the author's responses**

All my comments were addressed – although those involving extra analyses were unfortunately denied - so if not in this paper ... then hopefully in a future contribution. When the authors state as the goal "to evaluate aerosol optical depths for irradiance computations", then also the clear- sky LW fluxes need to addressed, especially near dust sources. Typical mineral dust aerosol radii near sources are 3-6um and even transported dust has effective radii of about 1.5um. Their given justification to ignore LW effects assumes too small (ca 0.5um) radii for mineral dust. In that context also the last sentence of the abstract ('... are unknown.') is very unsatisfying.

*We do not disagree with this reviewer's desire to investigate more fully the dependence of the SYN1deg irradiance calculations on aerosol optical depth and type. However, in the context of the paper, as written, it is difficult to extend the analyses to individual site locations without greatly increasing the length of the text.*

*Our argument to exclude discussion of LW irradiance in the paper (and in our first response) was based on three factors. 1. Errors in LW down due to AOD are less significant than errors due to water vapor and temperature profiles, 2. Past comparisons indicate GEOS5.4.1 atmospheric profiles (used SYN1deg radiative transfer calculations) capture total precipitable water (PW) and temperature profiles quite well, and 3. Differences between observed and calculated LW down at the surface are smaller than*

*errors predicted by differences in PW and surface air temperature. Thus, error due to AOD in LW down is small. However, we have added section 4.2 to briefly discuss LW down with respect to AOD and there include the reviewers desired comparison of fine and coarse mode AERONET AOD with MATCH equivalent aerosol types.*

For the dust inconsistency problem also possibly poor choices of AERONET references contribute, as (1) column properties (like AOD) of regions should not be compared to mountain site data (as Izana) and (2) months when other aerosol types dominate should be excluded (as for DJF in western Africa).

*Besides the increased length, individual sites may, or may not be representative of larger regions. (A point made by the reviewer in his third comment below.) We note too that we did not include the Izana site in the statistics given in the paper. However, we did indicate it as a site in the North Africa group in Appendix A. That is an error on our part and the site has been removed from the list in the table.*

And if AERONET is applied to reveal biases for AOD, then also the offered AOD split between fine-mode/coarse-mode AOD should be applied. I reject the notion that MATCH does not contain (approximately) this size information with its component processing, where dust and seasalt basically define the coarse mode AOD and (organic and black) carbon and sulfate define the fine-mode AOD. And for cases, where/when relevant AERONET sites are missing (as for northern Africa, central Africa or ocean regions), comparisons at coarser temporal (e.g., monthly, seasonal) resolutions (as offered by top down climatologies or other approaches such CERES, SRB 4, ICAP) are (as in my initial review) encouraged for insights on MATCH tendencies.

*We attempted to show in our first response the somewhat arbitrary definition of fine and coarse modes from AERONET being the result of a bimodal distribution, where the two modes can vary in particle size depending on the site location. However, to address the reviewers concern we have included a new section (4.2) to the paper to consider the impact of particle size on the SYN1deg irradiance calculations. We use the reviewer's suggested groupings of carbon and sulfates for small particles and large dust and sea salt for large.*

The strength of MATCH AOD data are high temporal resolution AOD maps (as needed in modeling) to include short-term regional anomalies, but that does not mean that global multi- annual (average) maps are correct. The most important section is chapter 4. So there should be more weight (and analysis) and less weight on the first 25 pages. In particular, the detailed comparisons to MERRA distract, as it is only a different model interpretation with quite different results and MERRA results are even left out in the comparisons at TOA fluxes (CERES) and surface fluxes (BRSN) of chapter 4.

*We have split section 4 into three parts, adding 4.2 "LW Comparisons". There we show LW irradiance calculations compared to observations along with a short discussion of the impact of particle size on the LW aerosol direct radiative effect at the surface and*

*incorporate canonical monthly means of coarse and fine mode AODs, both observed and from MATCH to show seasonality in both.*

On responses to my minor points of the initial review ...

- The demonstrated smaller AERONET AOD at cloud-free conditions at the CART/Bondville sites may be meteorology (air-mass) related. These statistics will not necessarily apply to many other AERONET sites, for instance at dust dominated sites where higher AOD will likely be associated with less clouds.

*While true the observations in the central United States cannot be applied across the globe, they should be representative of large agrarian and midlatitude plain regions that do represent a significant portion of northern hemisphere land types. We cannot include analyses at all the individual sites, however, we have included plots in Section 4.2 at three desert sites to more closely look at dust dominated areas.*

- The assumed 1um mineral dust size (0.5um eff radius) is very small, even for transported dust. Thus, the associated reasoning for "LW impacts are not important (to the surface irradiance)" is not convincing at all. Also with larger (up to 10um) dust sizes the effective solar absorption potential (e.g. 1-SSA) quickly increases so that for the same AOD the solar surface irradiances will be lower (dust size in MATCH could be possibly included via a proportional link between dust AOD and dust effective size, as sizes near dust sources are usually larger).

*We have divided Section 4 into 4.1 Shortwave and 4.2 Longwave discussions. In section 4.2 we address concerns regarding incorrect particle size and fine/coarse mode comparisons at desert African sites.*

- in Figure 1 there are not the promised maps for SSA and g ....?

*We added the spectral properties of the aerosol types used in the Langley Fu & Liou Radiative Transfer code. We did not add maps showing their distribution globally. (See our first response to Reviewer 1, question 5.)*

- I agree with the authors and retract from my initial conclusion of a "likely stronger fine-mode absorption in MATCH compared to MERRA", because Figure 4 compares model simulated AOD maps. Still, the larger MATCH AOD values in regions, where fine-mode aerosol (SU, OC, BC) types dominate are concerning – possibly helped by the fact that MATCH assimilations only constrains the total AOD and not local component mixtures.

*See section 4.2 for added discussion of fine/coarse mode aerosols with respect to AERONET and MATCH.*

- I still suggest in AOD assessments to separate fine and coarse mode AOD contributions when comparing to MERRA and AERONET. For AERONET inversions detailed (ambient) 22 bin size classes a separation at the 0.528um size-bin boundary is

recommended and in modeling the separation by components with combining BC/OC/SU (fine) and SS/DU (coarse). AERONET also offers simple fine-mode fractions via the AOD spectral dependence (and the SDA method) from direct attenuation data. The fine-mode effective radius can certainly vary (as demonstrated in a figure by the authors) but there are always a minimum at aerosol radii of about 0.5um (unless there is a major volcanic eruption with effective sulfate aerosol radii near 0.5um).

*In the new section 4.2 we have added a plot showing canonical monthly means of MATCH aerosol types (as suggested by this reviewer) compared to the fine mode and coarse mode aerosols determined at several west Africa AERONET sites.*

- if you use high altitude mountain AERONET sites (e.g. Izana) then you get underestimates in comparisons to regional averages (bad choice!). Also West-African sites (e.g. Ilorin) are biomass dominated in NH winter and dust dominated in NH summer, which complicates a type association unless a seasonal separation is done. You also present fine-mode and coarse mode distributions but I suspect that fine-mode is not dust (as fine-mode dust is secondary to coarse mode dust).

*We do not include the high altitude Izana site in our statistics. It was mistakenly included in the table in Appendix A1 and has been removed from the table.*

**Comments to the responses**

36 these maybe convective regions ... but a much more important element is that these are regions with fine-mode aerosol maxima (by wildfires and pollution).

45 make sure to pick regionally representative AERONET dust site (exclude mountain sites).

*The Izana site was not included in the original paper's statistics. However, Table A1 has it listed as a member of the North Africa group.  It has been removed from the Table A1. (The map of AERONET sites, Figure 7, shows the Cape Verde site which is stated as being 60m above sea level.)*

47 if AOD is correct, but the dust size (and solar absorption) is underestimated, then a model yields too high solar irradiances at the surface

*We have added a small table and discussion in the new Section 4.2 showing the dependence of SW and LW irradiance down at the surface on dust particle size.*

152 OPAC is parts in outdated (e.g. too much dust absorption, too little BC absorption) ... but more importantly, how is OPAC applied? I assume (if so state that) that MODIS AOD are compared to simulated AOD and differences are expressed via component mass corrections by applying OPAC aerosol type based Mass Extinction Efficiencies– assuming locally/monthly fixed aerosol component mixtures.

*Table 2 shows how the aerosol properties defined in the MATCH model are transferred to the Langley Fu & Liou radiative transfer code (LFLRT). The OPAC properties are defined (as tables in LFLRT) spectrally for the indicated constituents. MODIS aerosol optical depths are assimilated into the MATCH model using modeled mass and OPAC mass extinction/scattering coefficient. Then MATCH optical thickness is scaled by the ratio of MODIS/MATCH AOD. This scaled AOD is passed into the radiative transfer code and used as a weighting function for the properties defined in the LFLRT. Optical properties of aerosols in the radiative transfer modeling are also computed by those listed in Table 2.*

158 Sinyuk (2003) dust RFI-imag values are much better than those in OPAC ... also you may note that in his paper the coarse mode dust effective radius (in Table 1) is ca 1.5um (number mode radius of 0.63um and stddev of 1.72).

*We use the Sinyuk properties for dust in the Langley Fu & Liou Radiative Transfer model.*

182 as from AOD differences aerosol component mass is adjusted ... this of course requires mass extinction coefficients for aerosol type (which also include water uptake). Are these coefficients based on OPAC and if so have they been checked for realism?

*Yes, those are from OPAC, d'Almeida, and Sinyuk et al. We check "realism" and use d'Almeida and Sinyuk et al. for Sea salt and dust (please see Table 2).*

205

222

errors in the captions: "thickens" and "left)" and "right)" ... and eastern EU / west Asia

*Thank you, that error has been corrected.*

230
there is no plausible explanation why MERRA is so much smaller than MODIS and why MATCH over many land regions is even larger than MODIS (based on Figure 2 differences I am surprised to see in Figure 3 that the global average of MATCH is smaller than MODIS? Also, in Figure 4 clear and all-sky MATCH-MODIS differences are positive, which is inconsistent with Figure 3 means).

*We used MERRA-2 for all-sky AOD comparisons because MODIS can only provide clear-sky AOD. The values in Figure 2 are a bit deceptive. Looking closely at the color bar  one finds that the light green color is, in fact, slightly negative and light green dominates much of the globe. So, while the large positive (red and orange) biases catch one's eye, the slightly negative values dominate in the global average*

265 I suggest to replace 'convective' with 'fine-mode AOD maxima'

*The term "convective region" is a general term. We kept 'convective' in the revised version.*

283 Anomalous high AOD by MATCH at near overcast conditions (as shown in Figure 5) will not matter much for solar irradiances at the surface, when clouds reflect most of the solar energy... or? So for solar irradiance all-sky AODs (and 'averages') are less meaningful that clear- sky AODs.

*In the SYN1deg product we compute both clear-sky fluxes (cloud-removed) and cloudy sky (aerosol removed) to assess aerosol effect on clouds. Therefore, cloudy-sky AOD does matter.*

I do not see that "Match-Modis differences are smaller than Merra-Modis differences". And

*This is the case in the global mean values.*

340 Many aerosol types have, if not monthly, so at least seasonal maxima, so an evaluation to AERONET (in Table 2) on a seasonal basis would be much more insightful ... even better with a separation into fine-mode and coarse-mode AOD.

*This has been done at three desert sites in section 4.2.*

387 ... so what is the conclusion? Spell it out. MODIS AOD are likely overestimated without QAC so that MATCH is relatively high compared to MERRA. But this does not answer the question why MATCH is significantly larger than MODIS over many continental regions as shown in Figure 2.

*Added the statement ", likely increasing MATCH AOD overall." To the text in section 3.*

393 If MATCH (via MODIS) misses large aerosol events, why are AOD values over continents (incl. wildfire regions) so large?

*As described in the text, MATCH will not assimilate an episodic event until after it has been observed by MODIS. It does not necessarily miss the event but there can be a time lag depending on clouds. These episodic events are not considered to be the main reason why MATCH AOD values are large over continents.*

422 the AOD data presented in Figure 9 are maxima (if I read the captions correctly). Why are not median and interquartile-range values are presented instead? And is there a direct sample link between AOD and PV of just max statistics?

*The values are not 'maxima'. The AOD and PW values have been scaled to the largest value between observed and modeled values to place the points on the same plot and keep relative magnitudes between values accurate.*

426 explain 100% post and neg error in the captions. If the guess is 0.2 ... then at 100% the lower bound is 0.1 and the upper bound 0.4, which would give an asymmetry not shown in Figure 10. Also, it would be nice to show results for a more absorbing (dust or wildfire) aerosol, because the error max at sza near 0.5 will vanish and will more simply decrease with insolation. Still, I do not see the value of this figure as not even an AOD value is given in the captions (as usually AOD uncertainty wants to be translated into W/m2 irradiance error). Why do I have to read the text to get the info and with those definitions even negative AOD are possible.? Figures and captions should be self-explanatory! But do we need section 3.1 other than stating AOD error as function of aerosol type and SZA... all of which could be placed in a table.

*We have changed the legend on the plot to be more descriptive and rewritten the text in the figure caption.*

486 CERES detects SW (and LW) upward radiation (and not solar 'irradiances' as at the surface).

*Correct, irradiances at the surface are the result of radiative transfer calculations, not observations.*

493 Aside from AOD and precipitable water there are other knobs for tuning such as (aerosol absorption, aerosol (dust) size and aerosol elevation, surface albedo). Just tuning with global averages may provide globally the correct result for the wrong regional/seasonal regions. Thus, a regional evaluation is strongly encouraged. Later surface albedo changes are included ... so are these not elements in the global adjustments, as they seem to be higher everywhere in Table 4

*The adjustments discussed here are done for the CERES EBAF-surface product. We include the discussion as AOD is one of the variables that may be tuned in the process and thus adds insight into the aerosols coming in from the MATCH model. These adjustments are done regionally (1deg by 1deg) for a monthly time series and so indicate larger, gross characteristics of the data product.*

500 to increase the shortwave reflection in terms of aerosol properties also less aerosol absorption and a smaller aerosol type (relative more fine-mode AOD) helps

*We do not have the capability to adjust aerosol particle size and type independently in the tuning process, only total column AOD.*

508 MATCH AOD is already high at mid-latitudes ... even to MODIS, so an additional 0.02 AOD seems to go the wrong way.

*MATCH is higher than MERRA-2 over high/mid latitudes but comparisons with AERONET show near zero or slightly negative comparisons. It is within the context of the AERONET comparison that we make the statement that additional AOD (as found in the EBAF-surface tuning) is not unfounded.*

510 there is no 'top right' in Figure 10

*Thank you. The text should have referenced Figure 12 in the original manuscript (now Figure 13). It has been corrected.*

519 MODIS AOD over oceans are already likely too high, so that further increase cannot be justified.

*Adjustments through tuning are regional, not global, for the EBAF-surface product. We do not know the reason for the reviewer saying that MODIS AOD over oceans are too high.*

534 I miss in Table 4 the clear-sky upward MATCH based SW (and LW) uncorrected upward fluxes (to compare to CERES based TOA obs) and the clear-sky downward MATCH based SW (and LW) uncorrected downward fluxes (to compare to BSRN/buoy data).

*Table 4 (now Table 6) shows results for EBAF and EBAF-surface fluxes along with adjustments made during the EBAF-surface tuning process. It is not meant to be a validation table but to add insight into the AODs provided by the MATCH model. SW and LW irradiance bias/standard deviations at surface observation sites are found in figures 13 and 14 where we now cover the surface validation in sections 4.1 and 4.2.*
* * *
*Reviewer – 2*

Thank you for your response and please forgive my delay in re-reviewing. The authors have replied to each one of my comments, covering most of them, even though not as thoroughly as I'd like for some of them. The revisions to the manuscript were clearly as limited as possible, but passable. Please find below my comments on the revision, where "Points" refer to the point numbering from the authors' response to my earlier suggestions and "Lines" refer to the new version line numbers. I consider the unreferenced earlier "Points" as covered satisfactorily.

Point 20) Previously I suggested that l. 447 should read "...scale nearly linearly with AOD between...". Or maybe I was mistaken, and the initial meaning was that the scaling was linear with cos(sza)? Anyway, the authors said that it is fixed, but it is not really.

*The calculations scale nearly linearly with cos(sza) as the reviewer says. We have added "with cos(SZA) " to the text.*

Lines 472-473 and line 532: Now the biases changed to 3 and 15 W/m2

*Thank you for catching that. The numbers have been corrected to the values in Figure 13.*

Point 22) The sentence is not removed, even though the authors say it was.

*Thank you. That paragraph has been rewritten to reflect the flow of the paper.*

Point 26) I think that the authors' point would benefit from including parts of their response to my comment in the manuscript. Even though the authors' response has assuaged somewhat my concerns, I still think that their approach here is more of a sensitivity study than a rigorous proposal for AOD, albedo, water vapor corrections to their product. I am not convinced that possible errors in aerosol types should not be included in the adjustments. My suggestion here is to convey more strongly to the reader that these modifications are one possible solution between many and not the most probable one.

*First, we do not suggest the results of the tuning solution as a rigorous solution for the input parameters and so in a sense, the reviewer is correct, it does provide an idea of the sensitivity of the TOA irradiances to the inputs. Secondly, each tuning variable requires, for the Lagrange Multiplier (LM) solution, an estimate of the uncertainty of that parameter, essentially on a global scale. Aerosol types vary dramatically from region to region of the globe. This would greatly complicate the LM solution and it is also not clear that such estimates of aerosol type uncertainty exist.*

Point 28) Although the authors claim to have fixed this, it is still wrong. Not only the fig. number but I think they mean "left" instead of "right". Also, the bias now is 11 and not 12 W/m2

*Thank you, these errors have been corrected in this latest version.*

---

## Author Response (AR3)

*Once again, we wish to this reviewer for such interest and close reading of our paper. It is a little concerning that you state: "This time there were not responses to my latest comments…"We regret that you did not see, (receive), our point-by-point response to your previous comments. In fact, most of the changes and subsequent improvements regarding the discussion of the effect of dust particles on our results came from your review. None the less, we see that you have found the changes based on the comments below.*
* * *
Review of re-revised paper:

**Evaluation of aerosol optical depths and clear-sky radiative fluxes of the CERES Edition 4.1 SYN1deg data product *by D. Fillmore et al.**

**Highlights**

- - needed paper as MATCH aerosol data are an important element CERES products
- - now different larger dust sizes are included (good) ... but not correct SSAs (bad)

**Concerns**

- - solar flux validation for aerosol cannot be drawn from AOD data alone – aerosol absorption and aerosol size are also important.
- - apparent effect of decreasing SSA at increasing coarse dust sizes is ignored → wrong absorption is assumed

**General comments**

I have now reviewed the paper several times. This time there were no responses to my latest comments ... so I just re-read its content. I am delighted that now larger dust size are considered (great), but unfortunately the associated lower SSA (with larger sizes – for the same RFimag) are ignored. This becomes quite apparent from the supplied simulations in Table 6.
I have provided similar simulations, with the associated stronger AAOD (from lower SSA values) at larger sizes and I could demonstrate that for the particular set-up solar losses to the downward surface fluxes are twice as large with a dust size increase by a factor of 4. Thus, with a stronger mineral dust absorption, it is also likely that the' unresolved' issue for dust regions will disappear.

Thus, please redo your dust simulations and re-evaluate the dust area flux comparisons before publication.

*Based on your comments below regarding the dependence of irradiance on particle size, specifically Table 6 in the paper, we have found that one SW number was incorrect. Once the correct number was in place, we find that we do not need to re-evaluate our calculations as they agree with the values provided in your review. While, at the same time we also agree that the SSA in the OPAC tables being used for dust particles likely underestimate SW absorption. See below for discussion.*

**Specific comments**

37/38 Clarify this sentence: both Merra and Match are assimilations and the MODIS AOD input is the same. If not explain. Or has this to do that Merra allows for washout processes?

*Line 37/38. We have rewritten the sentence to be more precise to read:*

 *'The difference is largely due to the differing methods of assimilating the MODIS AOD data product and the use of quality flags in our assimilation.'*

39-42 I assume that refers to Match AOD. So, the surface-flux comparison indicates solar attenuation is too small (-> AOD underestimates and/or absorption underestimates

*Yes, those are the two primary possibilities.*

42-45 So the CERES clear-sky reflection needs more reflection of the surface site (-> AOD overestimates and/or absorption underestimates.

... so if we combine the two findings... then the only conclusion is that MATCH aerosol absorption is underestimated (which for dust is likely an underestimation in size)

*MATCH optical thicknesses over desert sites for clear- and all-sky conditions are larger (Tables 3 and 4). Computed downward shortwave from North Africa groups is larger than observed downward shortwave irradiance at the surface. As the reviewer suggested, it appears that we underestimate shortwave absorption, if particle size is larger. Here are the single scattering albedos as a function of dust particle size with OPAC refractive index. These single scattering albedos are low compared with those provided by the reviewer.*

*Single scattering albedo of dust particles with OPAC refractive index*

| Particle size | 1.9 (2.15) | 0.78 (2.0) | 0.39 (2.0) |
|---|---|---|---|
| 546 nm | 6.67931E-01 | 8.00413E-01 | 8.78470E-01 |
| 642 nm | 7.15135E-01 | 8.46123E-01 | 9.11902E-01 |
| 842 nm | 7.63865E-01 | 8.85041E-01 | 9.37848E-01 |
| 1226 nm | 7.84489E-01 | 9.02001E-01 | 9.47219E-01 |

We believe that the dust size the reviewer mentioned in his comment on line 174 is too large. But we get similar single scattering albedos with a smaller size using OPAC refractive indices.

46/47 leaving issues unexplained (even in the abstract) is discouraging 167 now several dust sizes are allowed (great !!)

*Line 47/49. We have changed the sentence to suggest the error may be due to the choice of dust particle size and distributions.*

174 the SSA (and ASY) varies with large dust size (and even spectrally), update Figure 1 and simulations! For mineral dust re of 1.5, 2.5, 4.0, 6.5 and 10um the mid-visible SSA values are 0.962, 0.931, 0.918, .882 and .840 for the same imaginary part (here 0.0011). In addition, since the dust solar spectral Rfimag are larger towards the UV the SSA value at shorter wavelengths are even lower (more absorbing)

*Figure 1 shows the values currently in the radiative transfer model and so represent the calculations in the SYN1deg data product. Changing the values is not currently an option for the operational SYN1deg product.*

223 just curious... are the land-sea contrast offsets of northern Africa a MODIS or a modeling problem? (I do not see very strong offsets in MODIS statistics.)

*Because a large land-ocean contrast over the west coast of north Africa does not appear on the right side of Figure 3, it appears to be a modeling problem.*

232 Merra apparently includes other AOD data from other sources (MISR, AERONET), but MODIS data should dominate (in volume) so I would not expect so significant differences as displayed in Figure 3. How can you exclude that model-specific aerosol processing in the base- line model (without assimilations) is not the issue. This could be easily verified ... or?

*We believe it is the MATCH model (and so 'base-line model assimilation') that is one of the primary sources of the differences. We state as much in lines 234-235.*

259 the least square fits and rms values (in figure 4) are dominated by the largest AOD, but not by the most frequent AOD, thus possibly also show the scatter plots in log/log scale

*The log density plot does show the vast majority of AODs are less than ~0.6. And though the fit line is 'pulled down' somewhat by the larger AODs we feel a log/log plot on top of the log density would not significantly change the results presented. It might bring the fit line a little closer to the 1-1 line, but the point that both MERRA2 and MATCH underestimate AOD relative to MODIS will not change. It is also re-iterated in the statistics below the plots.*

459 yes! ... and remember a larger coarse dust size can also increase the aerosol absorption

489 this table is interesting and quite revealing. To demonstrate, I did similar simulations, using a solar zenith angle of 0.95 (not 1, oh well but close), a desert surface albedo, a mid-lat summer atmosphere, and dust with (only) a mid-visible optical depth of 0.2 for mineral dust located between 1 and 3 km altitude. Here different dust sizes with their associated SSA are applied (all dust size-distributions assume the same spectral refractive indices with RFimag at 550nm = 0.0011). The downward shortwave and longwave fluxes of these simulations are

No dust Dust (0.2) Dust (0.2) Dust (0.2) Dust (0.2) Dust (0.2)

reff= 1.5um reff= 2.5um reff= 4.0um reff= 6.5um reff= 10.um

shortwave longwave

| 998 | 346 |
| 985 | 354 |
| 983 | 357 |
| 980 | 359 |
| 974 | 359 |
| 968 | 359 |

These calculations consider the lower SSA at larger size (as they should) and the solar losses at the surface flux losses for factor 4 dust size increase with these calculations are:
-15 W/m2 (between reff 10 and reff 2.5) while the author's simulations - even with larger u0 (=1) - 8 W/m2

(between reff 8 and reff 2). So please redo your calculations with the correct (lower) ssa values at larger sizes ... and you will have an explanation to your dust bias.

And the LW dust impact on downward fluxes depends strongly on the assumed dust vertical distribution as much as on size. Thus, all size (... from AeroNet inversions?), correct RFimag (especially in the stronger absorbing 8-10um region ... from I.Sokolik?) and altitude (...from Calipso?) have to all accurate for useful clear-sky dn LW flux comparisons at the surface.

So when the correct dust sizes/SSAs are applied with the result of a stronger aerosol absorption possibly (with the right dust sizes, and right dust altitudes) correction to AOD and water vapor may not be necessary to bring SW and LW fluxes into agreement.

*First. we'd like to point out that there was in fact an error in Table 6. The value for Downward SW irradiance for particle size 2.0$\mu m$ should have been 1038Wm$^{-2}$, not 1028Wm$^{-2}$. This has been corrected in the paper, line 489. This then brings our table values into line with the values presented above by the reviewer. Specifically, he records, for a fourfold increase in particle size, a decrease in DSI from 983Wm$^{-2}$ to 968Wm$^{-2}$ (~16Wm$^{-2}$ or -1.5%). With the correct value in Table 6, a fourfold increase in particle size (2.0$\mu m$ to 8$\mu m$) the DSI in our calculations decreases from 1038Wm$^{-2}$ to 1020Wm$^{-2}$ (~18Wm$^{-2}$ or 1.7%). If we change our cos(SZA) to 0.95 (as the reviewer used) the DSI values at 2.0$\mu m$ and 8$\mu m$ are 977Wm$^{-2}$ and 960Wm$^{-2}$. So again, a similar decrease of ~17Wm$^{-2}$ but the same percentage of 1.7%.*

*We thank the reviewer for pointing out the discrepancy which was in fact a mistake on our part. We also agree that the discrepancy between the SW down observation and calculations is likely based on the absorption characteristics of the dust particles (based on the SSA's in the table above). We cannot however, at this point in time, change the results in the SYN1deg data product.*

*The longwave numbers are correct in Table 6. To comment on the reviewer's statement that 'LW dust impact on downward fluxes depends strongly on the assumed dust vertical distribution as much as on size', in our offline radiative transfer model we increased the scale height of 1.5km to 5.0km for the same inputs as shown in Table 6. This resulted in DLI values at 0.5$\mu m$ , 2.0$\mu m$ and 8.0$\mu m$ particle sizes of 351Wm$^{-2}$, 357Wm$^{-2}$ and 359Wm$^{-2}$ respectively. Thus, increasing the scale height of the dust particles did not change the DLI by more than 1%, less than the changes due to particle size. This of course kept the particle size the same throughout the column which does not account for the fact that smaller dust particles are more likely to be lofted higher in the atmosphere.*

577 more clouds→more AOD? ... not for dust, when clouds remove dust

*In this case we imply that more clouds indicate more AOD in the MODIS AOD retrieval process as reported in the Varnai et al, 2017 reference.*

593 redo your large dust SSA calculations ... and the solar surface vs toa flux difference problem

for mineral dust should be gone.

*At this point in the processing of the MATCH model and its subsequent use in the SYN1deg radiative transfer calculations, we cannot 'redo our large dust SSA calculations'. What we have done, and at this reviewer's suggestion is try to show, in the paper, potential error due to the constraints we currently have on our dust models in the radiative transfer code (Table 6) and the MATCH model's ability to define large and small dust, Figures 15, 16 and related discussion.*

Below I attach a summary of the top-down (optics → component) approach of the MAC climatology, which lists in Table A1 different dust types, in Figure A3 (column3) seasonal averages of dust size (based

on AAODc and AODc-DU), assumed size-distributions and spectral refractive indices in Figure A6 and resulting single scattering properties in Figure A7 (where 'DU+' is for larger dust with reff =4um and 'DU' is for background dust with reff=1.5um).

*We thank the reviewer again for the information and the thorough reviews of the paper. We have added a sentence in the acknowledgements stating how the reviews have improved the paper overall, Lines 610-612.*

---

## Author Response (AR4)

Dear Dr Hatzianastassiou,                                                    15 July 2022

We appreciate this opportunity to address your concerns and those of the reviewer and submit a final version of our paper. Much of the discussion in this last review (as well as throughout the review process) has revolved around the treatment of large dust particles and possible resulting errors in downward surface irradiance. The treatment of dust in general has been an important theme of the reviewer and we have previously added, thanks to this reviewer's suggestions, Table 6 and Figures 1, 14, and 15 which discuss the impact of dust on the radiative transfer results. We also, previously, added a new LW section (4.2) and Figure 16 detailing differences between fine and coarse mode observations by AERONET and values supplied from. In the most recent comments, the reviewer has rightly pointed out that our treatment of large dust particles can lead to the errors in downward shortwave and longwave irradiances over deserts as found in our results. As this appears to be the final remaining significant concern, to clarify this point we have added comments in Section 2 (lines 174-178) and in the Conclusion (lines 598-605). We have also clarified that all dust particle sizes are in term of effective radius.

The reviewer's remaining comments (in italics), are, we feel, adequately addressed in the paper to date. We have collected them below for clarity and address each in turn.

*"Aside from the dust-issue though, there are still larger differences between MATCH and MERRA which at least should receive some attention – as both models assimilated MODIS AOD data, which would imply a better agreement than demonstrated. So, make sure that critical deficiencies and differences are well explained."*

And

*"It is a bit puzzling to me that you say that the 'undesired' land-sea contrast is likely associated with the MATCH model – as models are not expected to show these inconsistencies. It would be nice to get some more insights as to the reasons .. why?. "*

We disagree with the reviewer that the differences between the MATCH and MERRA2 AODs do not 'receive some attention.' Section 2 is dedicated to the differences found between MERRA2 and MATCH and reasons for such differences are discussed therein (for example see lines 218-220 and lines 277-283) and in various places throughout the paper. Other places discussing reasons for differences include the Abstract (lines 37-40), Section 2.3 (lines 336-337), lines (353-356) and in the Conclusion (lines 585-587). One topic he reiterates in this final review is the large negative values found in the clear sky comparisons of MERRA2 to MODIS over tropical oceans (Figure 3). While this bias does stand out and we mention it to the readers (lines 246-248) we do not have a good explanation so avoid making suppositions regarding another group's product.

*"Are there arguments why MATCH or why MERRA assimilations should be trusted more ... as the resulting AOD maps are so different? Does it simply mean that the influence of the baseline model (e.g. CAMS vs GOCART) is much stronger than the impact of the AOD data assimilations? That would be an important 'help/issue' to any manuscript reader."*

This is a good question but one that reaches beyond the scope of this paper. The goal of this paper is twofold, to analyze the output aerosol optical depth from the MATCH model against observations and MERRA2, and to evaluate the resulting surface irradiances given the knowledge found in those AOD comparisons. None the less, we suspect the answer is basically yes, the influence of the underlying model is greater than the influence of the assimilation of MODIS AODs. However, determining that rigorously is left to a separate study.

*"The biases vary with AOD ranges (usually satellite remote sensing suggest larger (and likely overestimates) AOD at low AOD values, while AOD maxima are often missed) so possibly a separate linear plot for just the 0-0.3 AOD range could be an useful extra."*

Again, we feel that information gleaned from such a plot is insufficient to warrant another plot and subsequent increase in the length of the paper.

In conclusion we believe we have addressed the reviewer's major concern regarding large dust particles and include new text in the paper and its conclusion discussing this concern. The reviewer's remaining comments, we believe, are either already adequately addressed within the text or are essentially a conversation on the strengths and weaknesses of aerosol assimilation and the models presented. This the reviewer points out with the comment "*I do want to not to hold up a publication any further.*"

Sincerely,

David Fillmore and co-authors.